# Biosensors, Artificial Intelligence Biosensors, False Results and Novel Future Perspectives

**DOI:** 10.3390/diagnostics15081037

**Published:** 2025-04-18

**Authors:** Georgios Goumas, Efthymia N. Vlachothanasi, Evangelos C. Fradelos, Dimitra S. Mouliou

**Affiliations:** 1School of Public Health, University of West Attica, 12243 Athens, Greece; ggoumas@uniwa.gr; 2Laboratory of Clinical Nursing, Department of Nursing, University of Thessaly Larissa, 41334 Larissa, Greece; efoula12@gmail.com (E.N.V.); efradelos@uth.gr (E.C.F.); 3Independent Researcher, 38500 Volos, Greece

**Keywords:** biosensors, Artificial Intelligence, Artificial Intelligence biosensors, AI, AI biosensors, false test results, false positive, false negative, false results, novel biosensor

## Abstract

Medical biosensors have set the basis of medical diagnostics, and Artificial Intelligence (AI) has boosted diagnostics to a great extent. However, false results are evident in every method, so it is crucial to identify the reasons behind a possible false result in order to control its occurrence. This is the first critical state-of-the-art review article to discuss all the commonly used biosensor types and the reasons that can give rise to potential false results. Furthermore, AI is discussed in parallel with biosensors and their misdiagnoses, and again some reasons for possible false results are discussed. Finally, an expert opinion with further future perspectives is presented based on general expert insights, in order for some false diagnostic results of biosensors and AI biosensors to be surpassed.

## 1. Introduction

Biosensors have gained widespread interest and acceptance as crucial and essential tools in clinical medicine. Nowadays, they provide rapid and precise detection of chemical and biological diagnostic markers and other markers for treatment monitoring. Such devices utilize biological recognition elements like certain proteins as well as nucleic acids in order to interact with determined targets and generate measurable signals [1]. Their capability to offer real-time results and their notable sensitivity have made them indispensable for the diagnosis of various diseases and several other applications including therapeutic drug monitoring [2]. Biosensors have boosted the efficacy of healthcare via allowing point-of-care testing and eliminating the need for time-intensive procedures.

Nowadays, Artificial Intelligence (AI) advancements are revolutionizing the modern lifestyle by enabling unprecedented automation, precision, and decision-making levels through technologies such as machine learning (ML), natural language processing, and computer vision; such achievements are evident in medicine, too. ML includes algorithms which enable computers to learn from information and improve over time, natural language processing allows computers to understand, interpret, and generate human language, and computer vision enables machines to analyze and interpret visual data from the world—including images or videos [3]. For example, AI has begun playing a significant role in enhancing biosensors, which are nowadays increasingly being integrated with AI to enhance their functionality. AI can process vast amounts of data by biosensors, boosting their sensitivity, enabling real-world analysis, and providing predictive insights, and this synergy among AI and biosensors has led to more precise and faster diagnostics [4]. The functionality of biosensors has been enhanced by AI algorithms, which can process complex biological information and recognize patterns, and they can also produce insights that would be difficult or impossible to be driven manually. These achievements have resulted in breakthroughs in various fields including oncology and cardiology, where AI-enabled biosensors can identify critical biomarkers and monitor conditions in real time. By combining the analytical abilities of AI with the inherent accuracy that biosensors can provide, such technologies are transforming clinical medicine and paving the way for a more personalized healthcare [4].

Nevertheless, the Coronavirus Disease 2019 (COVID-19) pandemic has highlighted that no diagnostic tool is infallible [5,6,7,8,9,10,11,12,13,14]. Furthermore, there are reports indicating that AI-powered biosensors are not always precise and accurate. False positives and false negatives can still occur, and, doubtlessly, these errors may have significant implications in clinical medicine. Such inaccuracies can arise from various causes, potentially leading to a range of adverse consequences [15]. Although these technologies possess advanced capabilities, AI-powered biosensors must be used with a thorough understanding of their limitations and should be employed in conjunction with other diagnostic methods to ensure accuracy and reliability in patient care.

Therefore, the aim of this novel and critical state-of-the-art review article is to present the common types of AI-enabled biosensors and the various factors that may influence their final test result. Additionally, future perspectives are discussed to help mitigate such incidents. To the best of our knowledge, this is the first comprehensive review addressing the potential for false results in both conventional and AI-based biosensors.

## 2. The Concept of “Biosensor” in Medicine

### 2.1. The Concept of “Sensor”

A sensor is a device or module designed to detect changes in physical quantities such as pressure, temperature, humidity, motion, force, or electrical parameters like current. A sensor converts these changes into signals that can be measured and analyzed. A transducer, on the other hand, is a device that facilitates the conversion of energy from one form to another [1]. Sensors are at the core of measurement systems. An ideal sensor exhibits characteristics such as a wide range, minimal drift, accurate calibration, high sensitivity, good selectivity, linearity, high resolution, reproducibility, repeatability, and quick response time. Advancements in sensor technology have become increasingly crucial due to its diverse applications. These applications include monitoring environmental and food quality, medical diagnosis and healthcare, automotive and industrial manufacturing, and areas like space exploration, defense, and security. Sensors are critical in diagnostics since they enable the precise and real-time measurement of divergent physical, chemical, and biological parameters, which are crucial for identifying, controlling, and estimating various health conditions. Sensors can be [1]:

(i) *Active and passive*, with the first to require an external energy source to function, such as microphones, thermistors, strain gauges, and capacitive or inductive sensors and they are known as parametric sensors, as their output depends on the measured parameter; passive sensors, on the other hand, generate their own signals without needing external energy, like thermocouples, piezoelectric sensors, and photodiodes and they are referred to as self-generating sensors;

(ii) *Contact and non-contact*, with the first to need to physically touch the stimulus in order to function, like temperature sensors, whereas the non-contact ones, however, can operate without any physical interaction, such as optical sensors, magnetic sensors, and infrared thermometers;

(iii) *Absolute and relative*, which can respond to a stimulus based on an absolute scale like thermistors and strain gauges, and measure a stimulus in relation to a fixed or variable reference point, such as a thermocouple that measures the temperature difference or pressure sensors that measure pressure relative to atmospheric levels, respectively;

(iv) *Analog and digital sensors*, with the first to convert a physical quantity into an analog form, which is continuous over time such as thermocouples, resistance temperature detectors and strain gauges, and the others to produce output as discrete pulses, with encoders being a common example;

(v) *Signal detection*, in which sensors can also be classified based on how they detect signals including physical sensors, chemical sensors, thermal sensors, and biological sensors.

### 2.2. The Common “Biosensor”

A biosensor is “a self-contained analytical device that combines a biological element (biosensing components) with a physicochemical component (biotransducer component) to generate a measurable signal for detection of an analyte of biological importance” and it is composed of the following five components [16,17,18,19]:

(i) *Analyte*, which is the specific substance being targeted for detection; for example, glucose serves as the analyte in biosensors designed to measure glucose levels, and they generally include endogenous or exogenous (drugs, etc.) substances like ions, small molecules, nucleic acids, proteins, viruses, bacteria, or even whole human cells that are typically biomarkers of real health or disease;

(ii) *Bioreceptors* are molecules that specifically interact with and recognize the analyte and examples include enzymes, cells, aptamers, DNA, and antibodies, while when a bioreceptor binds to the analyte, it generates a signal (such as light, heat, pH, electrical charge, or mass changes), a process referred to as biorecognition;

(iii) *Transducers* are devices that convert energy from one form to another and in the context of biosensors, they transform the biorecognition event into a quantifiable signal—this energy conversion process is called signalization—and most transducers generate optical or electrical signals, which are proportional to the interactions between the analyte and the bioreceptor;

(iv) *Electronics* are segments that process the signals produced by the transducer and prepare them for display, including sophisticated circuitry for signal conditioning, such as amplification and the conversion of analogue signals into digital form and the refined signals are then quantified and relayed to the biosensor’s display unit;

(v) *Displays* are the interfaces that present the results in a format comprehensible to the user and may consist of devices like a liquid crystal display or a direct printer that outputs results as numbers, graphs, tables, or images, with a system to combine hardware and software to deliver user-friendly outputs based on the biosensor’s data.

The origins of biosensors can be traced back to 1906, when M. Cremer demonstrated that the concentration of an acid in a liquid was directly related to the electric potential generated between different parts of the fluid separated by a glass membrane [20]. In 1909, Søren Peder Lauritz Sørensen introduced the concept of pH, representing hydrogen ion concentration. In 1922, W.S. Hughes developed a pH-measuring electrode [21]. During the period between 1909 and 1922, Griffin and Nelson demonstrated the immobilization of the enzyme invertase on aluminum hydroxide and charcoal, marking another important step in biosensor development [22,23]. The first true biosensor, an oxygen detection device, was created in 1956 by Leland C. Clark, Jr., who is widely regarded as the “father of biosensors”. His invention, the “Clark electrode”, laid the foundation for modern biosensor technology, yet the term “biosensor” was initially introduced by Cammann in 1977 [24]. In 1962, Leland Clark introduced an amperometric enzyme electrode for glucose detection. In 1969, Guilbault and Montalvo, Jr. developed the first potentiometric biosensor for urea detection. By 1975, Yellow Spring Instruments (YSI) introduced the first commercial biosensor, ushering in the era of practical applications for these devices [25]. Since then, the field of biosensors has evolved into a multidisciplinary research area that integrates principles from physics, chemistry, biology, micro/nano-technology, electronics, and medicine. Following the development of the i-STAT sensor, remarkable advancements have been achieved. From 2005 to 2015, the database “Web of Science” indexed over 84,000 reports on biosensors, reflecting the extensive global research and innovations in this dynamic field.

The development of biosensors is categorized into three generations, based on how the biorecognition element (bioreceptor) is integrated with the transducer. In the first generation (1st generation), biosensors detect the analyte content and the products resulting from the bioreceptor’s reactions, which then diffuse to the transducer’s surface, generating an electric response. These sensors are also known as mediator-less amperometric biosensors. Leland Charles Clark Jr., often referred to as the father of biosensors, first outlined the components of a biosensor in a 1956 report, which introduced an electrode designed to measure oxygen concentration in blood [26]. In the second generation (2nd generation), additional components such as auxiliary enzymes and co-reactants (which can include artificial or partially toxic mediators or nanomaterials) are incorporated into the biological component layer of the biosensor to improve its analytical performance. These sensors are referred to as mediator amperometric biosensors. In 1976, Clemens and colleagues introduced an electrochemical glucose biosensor as part of a “bedside artificial pancreas” [27]. The third generation (3rd generation) represents a significant advancement, where the bioreceptor molecule is directly integrated into the core sensing element. This shift eliminates the reliance on freely diffusing mediators in the electrolyte. This generation focuses on establishing a direct interaction between the enzymes and the electrode, enabling electron transfer without the need for intermediary stages, such as those involving nanomaterials. Noteworthy advantages of this generation include reduced design costs and enhanced efficiency in performing repeated measurements [28]. Table 1 summarizes the generations of biosensors and their comparison.

Furthermore, biosensors possess some basic characteristics which reflect their performance [1,17]:

(i) *Selectivity*, which is an essential property when choosing a bioreceptor for a biosensor, allows the bioreceptor to identify and detect a specific target analyte in a sample that may contain various other substances and impurities;

(ii) *Sensitivity*, which refers to the biosensor’s ability to detect and identify even trace amounts of the analyte in low concentrations, it indicates the minimum detectable quantity required to confirm the presence of the analyte in a sample;

(iii) *Linearity* is crucial for ensuring accurate measurement results and higher levels improve the sensor’s ability to detect varying concentrations of the analyte accurately;

(iv) *Response time* is the duration required for the biosensor to provide 95% of its final results, reflecting how quickly it can respond to a change in analyte concentration;

(v) *Reproducibility* refers to the biosensor’s ability to consistently produce the same output when a sample is measured multiple times, and it also encompasses both precision and accuracy;

(vi) *Stability* is a critical feature in biosensors, especially for continuous monitoring applications, and it measures the sensor’s resistance to environmental factors that could affect its performance over time, such as the degradation of the bioreceptor or fluctuations in the bioreceptor’s binding affinity for the analyte.

As previously mentioned, bioreceptors play a fundamental role in the construction of biosensors. Depending on the type of bioreceptor used, biosensors can be categorized into several classes: enzymatic biosensors (the most common type), immunosensors (known for their high specificity and sensitivity, particularly useful in diagnostics), aptamer or nucleic acid-based biosensors (which are highly specific for microbial strains and nucleic acid-containing analytes), and microbial or whole-cell biosensors. Another way to classify biosensors is based on the type of transducer used. In this classification, sensors are divided into electrochemical sensors (further subdivided into potentiometric, amperometric, impedance, and conductometric sensors), electronic biosensors, thermal biosensors, optical sensors, and mass-based (or gravimetric) sensors. Additionally, some biosensors are classified according to the combination of bioreceptor and analyte, though this classification is more limited. Other classifications are based on the detection system (which can be optical, electrical, electronic, thermal, mechanical, or magnetic) and the technology used (including nanotechnology, Surface Plasmon Resonance (SPR), biosensors-on-chip, electrometers, and deployable biosensors) [1,16,17,18,19].

In healthcare, biosensors have various uses, such as detection of diseases, retinal prostheses, enzyme biosensing, phenotypic cancer diagnostics, MRI contrast, rapid nucleic acid diagnostics, diagnosis of health conditions, optical DNA diagnosis, overall health monitoring and medical mycology [16,17]. It should be noted that, apart from healthcare settings, biosensors are widely applied in various fields including environmental monitoring, food safety, agriculture, industrial and other applications. In environmental monitoring, they are used to detect pollutants and contaminants in air, water, and soil. In food safety, biosensors help in detecting pathogens, allergens, and spoilage microorganisms. In agriculture, they are utilized for monitoring soil health, crop conditions, and detecting pesticides. In addition, biosensors are employed in industrial settings to measure parameters such as pressure, temperature, and chemical composition. This improves efficiency and safety in manufacturing processes [2,29,30].

### 2.3. The Biosensors with Integrated Artificial Intelligence (AI-Enabled Biosensors or AI Biosensors)

#### 2.3.1. The History of Artificial Intelligence

Enhancing healthcare and overall quality of life has been identified as a top priority in the “America’s Artificial Intelligence for the Next 20 Years Research Roadmap”, and similarly, China’s “Healthy China 2030” initiative aims to advance wearable devices, smart health electronics, and mobile healthcare applications [4]. With the continual refinement of AI technologies and their integration with the Internet of Things (IoT), the field of intelligent biosensors is poised for rapid and transformative growth [4].

The term “Artificial Intelligence” was first introduced during a summer research initiative at Dartmouth in 1956 and it encompasses the creation of computational methods capable of matching or even surpassing human intelligence [31]. While some aspects of AI remain largely theoretical, advancements in fields such as machine learning are steadily transforming these ideas into practical applications. The recent rise of generative AI technologies has demonstrated the immense potential of AI, fueling unprecedented interest in the field. In the realm of science, AI excels in uncovering patterns and trends within massive datasets—an arduous task for the human mind. This capability enables the optimization of protocols, accurate interpretation of results, and the prediction of outcomes with exceptional precision [32]. In recent years, there has been an increasing focus on integrating intelligence into various systems, whereas the advanced processing of biosensing data has transformed biosensors by adding some more generations, whereas the fusion of AI with biosensors has created the interdisciplinary field of AI biosensors [4].

#### 2.3.2. Artificial Intelligence Applications in Biosensors

When discussing AI in the context of biosensing or other scientific disciplines, the focus often shifts immediately to its role in data analysis; while AI’s capabilities in processing and analyzing data are undeniably significant, they represent just a small portion of its full potential. AI techniques can be integrated into every stage of biosensor development, from selecting target analytes and designing recognition elements to improving signal transduction and interpreting results, including [32]:

(i) *Analyte selection*, where AI is set to improve biosensing by enabling the exploration of biomolecular pathways through omics-based approaches, discovering new biomarkers, and combinatorial multi-analyte analysis;

(ii) *Recognition element selection* due to the rapid evolution of various diseases and the rise of multifarious threats which have accelerated their discovery, driven by critical advancements like affordable and fast DNA synthesis and deep sequencing and evolution of advanced bioinformatics tools, and the introduction of sophisticated high-throughput experimental methods for functional characterization, and in this manner, the integration of AI in the identification of recognition elements is a further transformation, enhancing screening techniques for naturally occurring bioreceptors and even aiding in the design of totally new receptors via computational approaches;

(iii) *Transduction*, in which efforts in AI are centered around the smart development of new materials with improved biosensing features, the downsizing of current sensing equipment, and the creation of biosensing techniques that go beyond conventional biomolecular interactions;

(iv) *Data handling*, where AI techniques aim to produce accurate, high-quality data that offer reliable insights for biomolecular analysis.

As a result, the fundamental structure of AI biosensors consists of three key components [4]:

(i) *Data collection*, which involves biosensors that monitor physical, chemical, biological, environmental, or identity-related information continuously;

(ii) A *signal conversion system* that converts the collected data into an electrical output signal with specific sensitivity;

(iii) *AI-driven data processing*, which is divided into four stages: interface (i.e., wearable devices), data classification (i.e., diagnostic imaging), data modeling (i.e., climate change modelling) and analysis, and decision making (i.e., autonomous vehicles).

Flexible bioelectronic materials—including films, textiles, bandages, patches, and tattoos—serve as mechanical frameworks for embedding electronic circuits, which are essential components of AI-powered biosensors. Polyimide (PI), polyethylene terephthalate (PET) and polydimethylsiloxane (PDMS) are among the most common materials. Additionally, certain nanomaterials, particularly carbon-based ones, show great promise for flexible bioelectronics (also known as organic electronics) which hold the potential to enable a wide range of applications [4]. AI-powered biosensors are ideally integrated with wireless data communication systems, enabling seamless transmission of information between biosensors and smartphone-based platforms or other intelligent devices. Wireless AI biosensors eliminate the need for cables, reducing costs while offering scalability and adaptability for adding new devices. Key wireless technologies facilitating the widespread adoption of AI biosensor networks include Bluetooth, radio-frequency identification, and others [30].

Moreover, various machine learning algorithms—including support vector machines, principal components analysis, hierarchical cluster analysis, artificial neural networks and decision trees—have been used in AI biosensors for efficacious and precise decision making; SVMs are supervised ML techniques designed to identify the optimal hyperplane that separates two data classes, effectively addressing binary classification challenges [4,33]. Smartphone-based platforms are also becoming increasingly important in AI-biosensing, due to their integration of various sensors and functionalities such as data processing, communication, storage, and cloud interaction. While these platforms are capable of handling data acquisition, processing, and decision-making tasks similar to computers, they are primarily designed for consumer entertainment and possess inherent limitations [34]. Nevertheless, as it will be discussed in this review, all categories of AI biosensors face specific limitations and challenges that may lead to misleading results.

ML has been employed to enhance the specificity, sensitivity, and LOD of bioreceptor-free biosensors, extending beyond traditional chemical sensor arrays. ML has been integrated across a variety of biosensing mechanisms, with notable examples including imaging-based biosensors and SERS-based biosensors. Moreover, the application of ML in biosensors is not limited to bioreceptor-free designs. Several studies have highlighted instances in which conventional biosensors incorporate ML techniques to optimize their performance. ML is a subcategory of AI that enables computers to learn from information and further predict or decide with no explicit programming. ML can be widely categorized into [28]:

(i) Supervised learning, in which a model is trained on labeled data—meaning the input data are paired with corresponding correct outputs—thus allowing the model to learn patterns and establish precise predictions; this type of machine learning is further subdivided into classification (random forest, k-nearest neighbor and naïve Bayes) and regression (linear, Lasso and Ridge regression), and

(ii) Unsupervised learning, which is further subdivided into clustering (hierarchical, k-means and Gaussian mixture clustering) and dimensionality reduction (principal component analysis), and overall, a model is trained on unlabeled data, thus allowing it to detect patterns, structures, or relationships within the data without predefined outputs.

Further, deep learning—a specific subcategory of ML—utilizes neural networks to process complex, high-dimensional data with minimal human intervention. Nevertheless, despite their potential, ML models must be rigorously trained and validated to address issues such as overfitting, underfitting, and bias. This is essential to ensure accurate, reliable and generalizable performance [4,32].

## 3. Current Common Types of Diagnostic Biosensors and False Results

Various classification schemes for biosensors have been proposed. As previously discussed, in addition to the type of bioreceptor used, biosensors can also be classified based on their transduction mechanism, detection system, and underlying technology [16,17,18,19,35,36]. So, there have been discussed some relatively common categories, based on their:

(i) Bioreceptors, a category which includes the enzyme-, antibody-, aptamer-based, whole-cell-, nucleic acid-, and hormone-based biosensors and nanobiosensors;

(ii) Transducers, including the electrochemical (amperometric, potentiometric, volatmetric and conductometric biosensors), the optical, the electronic, the thermal, the gravimetric, the piezoelectric, the magnetic and the acoustic biosensors;

(ii) Detection system, which has similarities with the previous classification style and includes optical, electrical, electronic, thermal, magnetic and mechanical biosensors;

(iv) Technology, in which nanobiosensors, SPR biosensors, biosensors-on-chip, miniaturized biosensors and electrometers are included.

These categories will be discussed in short below.

### 3.1. Current Common Biosensors Based on Their Bioreceptors

Biosensors can be categorized as either catalytic or non-catalytic, depending on their biorecognition mechanism. In the catalytic category, the interaction between the analyte and the bioreceptor initiates a specific biochemical reaction, resulting in the production of distinct reaction products. This group includes biosensors, which employ tissues, whole cells, bacteria, or enzymes [36]. In contrast, non-catalytic biosensors involve the analyte binding irreversibly to the receptor without inducing a biochemical reaction. These biosensors rely on interactions with nucleic acids, antibodies, or cell receptors to detect their target analytes [36].

#### 3.1.1. Enzyme-Based Biosensors

Enzymes act as biocatalysts, which significantly accelerate various biochemical reactions within biological systems, and various approaches are employed to recognize analytes using enzymes [37]:

(i) The enzyme metabolizes the analyte, and the concentration is determined by measuring the catalytic conversion of the analyte;

(ii) The analyte inhibits or activates the enzyme, with its concentration linked to the alteration in the enzymatic product formation;

(iii) Changes in enzyme characteristics are monitored during the reaction.

Enzyme-based biosensors consistently dominate the scientific literature in terms of publication volume, followed by antibody-based biosensors. Enzyme immobilization techniques are of critical importance due to the inherent instability of free enzymes, difficulty in recovering their active forms, and other associated limitations [38]. However, immobilized enzymes present several advantages, including reusability, continuous operation and enhanced stability, while still maintaining substantial catalytic activity compared to free enzymes. Although they may exhibit a slower catalytic rate and require additional processing steps, immobilized enzymes remain widely employed in both medical and industrial applications. Their benefits include rapid regulation via enzyme removal, simplified separation from the final product, and improved long-term stability. Common enzyme immobilization methods include adsorption, covalent bonding, entrapment, and cross-linking [38].

Electrochemical biosensors are among the most widely utilized biosensors, operating based on the electrochemical characteristics of both the transducers and the analytes. It has been discussed that enzyme-based amperometric biosensors can be utilized in the following ways [39]:

(i) As “offline” devices, which involve collecting biological samples and analyzing target analytes using biosensor-equipped analytical instruments, for instance, commercially available devices are commonly used to measure blood glucose levels.

(ii) As “in vivo” sensors, which are implanted directly into the body to continuously monitor extracellular fluctuations in the concentrations of specific analytes, yet their use is largely restricted to preclinical research in animal models due to the invasive nature of these devices.

(iii) As “online” devices, which integrate biosensors with sampling devices implanted in the body or biological materials; for example, microdialysis probes can be inserted and linked to a flow-through detector containing a biosensor component.

In enzyme-based amperometric biosensors, the measured signal is the electrical current generated by the oxidation or reduction in electroactive species at the working electrode, which is typically made of materials such as gold, carbon, or platinum. The magnitude of this current is directly proportional to the analyte concentration in the test solution following the addition of the substrate. This current is recorded when a fixed potential is applied between two electrodes. Enzyme-based amperometric biosensors have been widely investigated due to their advantages, such as ease of miniaturization, robustness, and the ability to function with small sample volumes, even in complex biological or environmental matrices [38].

Amperometric enzyme biosensors are generally classified into three main generations based on the electron transfer mechanism employed to monitor the biochemical reaction, or the degree of integration between biosensor components such as the transducer, enzyme, mediators, and cofactors. Regardless of the generation, the presence of an enzyme is essential, and the performance of the sensor depends on multiple factors, including the operating pH and temperature [39]. First-generation biosensors—also referred as mediator-free amperometric biosensors—detect the concentration of analytes or the products of enzymatic reactions that diffuse to the surface of the transducer, producing an electrical signal [40]. Second-generation biosensors, also known as mediator-based amperometric biosensors, utilize mediators as oxidizing agents to serve as electron carriers, and this strategy enables operation at low potentials, thereby eliminating dependence on oxygen and minimizing interference from other molecules. Some commonly used mediators in this generation include ferricyanide and ferrocene [41]. A third-generation biosensor typically consists of three main components: the enzyme as the biorecognition element, a redox polymer (or nano-scale wiring element) to facilitate signal transmission, and the electrode serving as the entrapping surface. Incorporating a redox polymer to directly connect the redox center of the sensing enzyme to the electrode surface significantly enhances biosensor performance [42]. Although third-generation biosensors are still under development and not yet widely adopted for analytical applications, ongoing advancements in polymer science and nanotechnology render them highly promising. They are expected to exhibit rapid response times and reduced dependence on oxygen or cofactor concentrations. Today, a variety of commercial enzyme-based amperometric biosensors are available for measuring analytes such as glucose, lactate, and alcohol. These biosensors utilize oxidases, such as glucose oxidase, lactate oxidase, and alcohol oxidase—to oxidize their respective substrates, generating hydrogen peroxide (H_2_O_2_), which is subsequently detected by the electrode.

Since many enzyme reactions involve the release or absorption of hydrogen ions, which alter the ionic concentration of the surrounding medium, ion-selective electrodes can be employed to monitor these processes. In a potentiometric biosensor, the signal is generated by the potential (voltage) difference between the working electrode and the reference electrode, which is measured under equilibrium conditions—that is, when no current is flowing—to prevent interference with the enzymatic reaction. The measured potential is logarithmically related to the concentration of the target analyte, allowing for quantitative analysis [43]. As will be discussed in the next subsection, potentiometric biosensors are classified into the ion-selective electrodes (ISEs), the Enzyme Field-Effect Transistors (EnFET), and the Light-Addressable Potentiometric Sensors (LAPS), and it this paragraph, only the second category will be discussed. An EnFET sensor is designed using an Ion-Sensitive Field-Effect Transistor (ISFET), which is created by replacing the metal gate of a conventional Metal Oxide Semiconductor Field-Effect Transistor (MOSFET) with a reference electrode placed in an aqueous solution, and the gate is separated from the solution by the gate oxide layer, and an enzymatic membrane is positioned between the reference electrode and the gate oxide. This setup allows the EnFET to respond to changes in the concentration of specific ions, facilitating the detection of enzymatic reactions [44]. The EnFET device can be applied for both quantitative and qualitative analysis of enzyme–substrate reactions. These enzymatic reactions affect the accumulation of charge carriers on the gate surface, with the amount of accumulated charge being directly proportional to the concentration of the analyte in the sample [44].

Furthermore, enzymatic reactions often result in alterations to the ionic concentration, which in turn affect the electrical conductivity of the electrolyte solution. This alteration can be measured by a conductometric biosensor, which applies a potential difference between two parallel electrodes. As a result, ion mobility increases, with negatively charged ions moving toward the anode and positively charged ions toward the cathode. The conductivity of the solution is determined by the concentration and mobility of the ions, thus making this method especially useful when there are minimal or no electrochemical reactions occurring on the electrodes. Similar to other electrochemical biosensors, enzyme immobilization techniques used in amperometric biosensors can also be applied to conductometric transducers [43]. For instance, enzymes can be immobilized on the electrode surface within an albumin gel film through covalent bonding with glutaraldehyde, or using methods such as sol–gel entrapment, covalent binding to a collagen membrane, electrochemical polymerization, or cross-linking with bovine serum albumin via glutaraldehyde [43].

Moreover, in an electrochemical impedimetric biosensor, the measurable response is the impedance of the electrode. Electrochemical impedance spectroscopy (EIS) is employed to investigate changes in the interfacial properties resulting from biorecognition events that occur at modified surfaces. The resulting impedance spectrum can then be analyzed to derive quantitative insights into the electrochemical processes [45]. However, this impedance measurement method is less commonly used in enzyme-based biosensors when compared to potentiometric and amperometric techniques, primarily due to the time required to record a complete impedance spectrum across a wide frequency range. Additionally, the EIS technique demands specific conditions, such as linearity, stability, and causality, to ensure the validity of the impedance spectrum. As a result, EIS is typically employed as a characterization technique for most enzyme-based impedimetric biosensors [45].

Enzyme-based optical biosensors have been under continuous development over the past several decades. As it will be discussed in following subsections, these optical transducers detect variations in optical properties—such as fluorescence intensity, light absorption, reflectance, chemiluminescence, evanescent wave, refractive index, and Raman scattering—which arise from the interaction between a biocatalyst and the target analyte [46]. In a light absorbance-based enzyme biosensor, biochemical reactions occurring at the transducer surface alter the chemical environment. These changes, in turn, modify the light absorption characteristics of the biorecognition element at specific wavelengths. Typically, a single optical fiber or a fiber bundle is used to direct light toward the analyte-catalyst transducer surface. During the enzymatic reactions, the transmitted or reflected light is guided back to the detector through the fibers, where it is measured as a signal to monitor changes induced by the recognition element, the analyte, or the enzymatic products. For example, an absorbance-based biosensor for the quantification of glucose was developed using two enzymes—GOx and horseradish peroxidase (HRP), and the enzyme was entrapped in a polyacrylamide gel to create the sensor [47].

Many enzyme-based optical biosensors depend on detecting fluorescent signals from enzymatic reactions [48]. The first approach in developing such biosensors utilizes the intrinsic fluorescent properties of the enzyme itself. All enzymes exhibit fluorescence in the UV region of the electromagnetic spectrum due to the presence of three fluorescent amino acids—phenylalanine, tyrosine, and tryptophan—in their structures. The second approach utilizes the intrinsic fluorescence properties of enzymatic reaction cofactors to detect changes in fluorescence during the catalytic process [49]. Furthermore, luminescence arises when an excited molecule emits light as it returns to its ground state, with the emitted light having a longer wavelength and lower energy than the absorbed light; in an enzymatic reaction, the presence of molecular oxygen can quench photoluminescence, and this happens via a radiationless deactivation process, wherein oxygen interacts with the fluorophore, resulting in reduced light emission [50]. An innovative approach for enzyme-based optical transducers involves using the SPR technique [51].

In addition, nearly all enzymatic reactions are exothermic, releasing heat, which can be used to measure reaction rates and analyte concentrations. The total heat produced (Q) is directly proportional to the molar enthalpy change (∆H) and the total moles of product molecules (nP). A thermistor is a type of resistor whose resistance varies with temperature. A thermistor detects temperature changes—caused by the enthalpy change (∆H) of a reaction—and converts them into electrical signals such as resistance. Thermistor-based calorimeters, also known as enzyme thermistors (ET), utilize thermistors to monitor electrical changes that result from temperature variations during biocatalytic reactions, making them particularly suitable for quantitative analysis [52]. The ET functions by immobilizing the enzyme in a packed bed column maintained at a constant temperature. When the substrate flows through the column, it reacts to form a product, releasing heat. The temperature difference between the substrate and product is measured by thermistors at the column’s entry and exit points. These highly sensitive thermistors are capable of detecting even minor temperature changes, enabling precise thermal biosensing. The amount of heat released is directly proportional to the amount of substrate consumed during the reaction. Enzyme thermistors offer direct and interference-free measurements, typically without the need for additional reagents. They also allow for high enzyme loading, strong operational stability and extended shelf life. Additionally, ETs can be integrated into a Flow Injection Analysis (FIA) system for improved reproducibility, high throughput, and continuous analysis [52].

TELISA (Thermometric Enzyme-Linked Immunosorbent Assay) is a technique similar to ELISA, but instead of measuring an optical signal, it quantifies the heat produced by the enzyme label using an enzyme thermistor. There are two main types of TELISA: the direct competitive immunoassay and the sandwich immunoassay, following the same fundamental principles as traditional ELISA [53].

Finally, enzyme-based biosensors can be piezoelectric as well; the most widely used type of piezoelectric biosensor is the quartz crystal microbalance (QCM) [54], which will be discussed in the next subsection.

It should be noted that approaches for improving the utility and the overall performance of enzyme-based biosensors have been illustrated, and the main approach is to enhance the enzyme–substrate affinity by improving the substrate’s access to the enzyme’s active site. In enzyme engineering, this can be achieved through methods like [43]:

(i) Site-directed mutagenesis, which includes substitution or/and removal of enzyme amino acids or/and incorporation of non-natural amino acids and enzymatic addition of a genetic tag;

(ii) Protein fusion, which includes chemical modifications like site-specific modifications, non-specific alterations of the surface of the enzyme, chemical cross-linking, and the usage of polymers;

(iii) Multi-enzyme systems, where the principle behind these systems relies on enzyme cascade reactions, involving a series of consecutive biocatalytic steps.

It has been previously discussed that enzyme-based biosensors face several inherent limitations, including low specificity due to sensitivity to unrelated compounds, lengthy incubation times, and reliance on inhibition mode. This mode of operation necessitates both baseline testing and subsequent retesting after sample exposure, which can be time-consuming and cumbersome. Additionally, the irreversible nature of enzyme–ligand interactions prevents sensor reuse without a regeneration process, which is often inefficient and labor intensive [55]. A recent study also identified that the false-positive results in enzyme biosensor arises from crosstalk. In such cases, small molecules produced by high-producer cells into low-producer cells, activating the biosensor in these cells. This unintended activation can lead to the overrepresentation of false-positive or cheater cells in the enriched population. These cells do not genuinely exhibit high enzyme activity, but instead appear to do so due to the non-specific biosensor activation caused by molecular leakage [56]. Specifically, the sensitivity of enzyme inhibition-based biosensors is influenced by several key parameters. One of the most significant factors is the measurement protocol, as different approaches can impact the accuracy and reproducibility of the results. Immobilization methods, which determine how the enzyme is affixed to the sensor surface, also play a pivotal role, as they impact the stability and catalytic activity of the enzyme. Enzyme loading—the quantity of enzyme immobilized on the sensor—directly affects the signal intensity and the overall responsiveness of the sensor. Additionally, incubation time, defined as the period during which the enzyme interacts with the inhibitor, is critical for achieving optimal inhibition and ensuring accurate detection. Lastly, the substrate concentration used to measure enzymatic activity must be carefully optimized to ensure precise monitoring without saturating or underutilizing the enzyme’s catalytic capacity [57]. Apart from enzyme inhibition-based biosensors, the enzyme inhibition in general can result in false results. This can occur due to non-specific inhibition by contaminants or unrelated compounds or undesirable aggregations, reversible inhibition causing fluctuating signals, and irreversible inhibition that prevents enzyme reuse without regeneration. Baseline drift, interference by structural analogs, and environmental factors like pH or temperature fluctuations can significantly distort the accurate measurement of analyte concentration. Moreover, incomplete signal recovery after inhibition may result in false negatives, while analogs or similar molecules can produce false positives [58]. Therefore, proper controls, optimized assay conditions, and improved enzyme design are essential to mitigate these issues and enhance biosensor reliability. Therefore, enzyme-based biosensors are hampered due to problems concerning their solvent tolerance, pH and temperature stability [59,60,61]. Cross-reactivity with non-target analytes can lead to false positives, while loss of enzyme activity due to denaturation, degradation, or unfavorable environmental conditions, such as pH or temperature fluctuations, may also cause false results. Enzyme saturation at high substrate concentrations can limit signal production, resulting in underestimation of targets, whereas non-specific inhibitors in the sample can suppress activity, also leading to false negatives. Additionally, activation by unintended molecules, autolysis, or self-degradation of enzymes can generate erroneous signals [60,61,62]. Misfolded enzymes or improper post-translational modifications impair function, while residual activity after inhibition may yield false positives [61,63]. Additionally, the reliance on specific cofactors—whether deficient or present in excess—can distort enzymatic performance, leading to inconsistent or misleading results [61,63,64]. As a result, it is obvious that enzyme-related issues that can lead to false positives and false negatives in enzyme-based biosensors like enzyme instability due to denaturation or loss of active sites, or harsh storage conditions, batch-to-batch variability influencing sensitivity, and inhibition by matrix components. Structural changes from environmental stress, poor enzyme immobilization triggering leaching, and non-optimal enzyme loading can disrupt performance. Competitive binding of the target analyte by co-existing analytes, allosteric effects from non-target molecules, and unintended side reactions can compromise specificity. In addition, enzyme aggregation, pH sensitivity, oxidation, or chemical modifications may impair activity, while variations in enzyme concentration or loss of function due to weak binding further impact reliability. These factors necessitate rigorous enzyme characterization and sensor optimization for accurate biosensing.

#### 3.1.2. Antibody-Based Biosensors (Immunosensors)

Antibodies—whether naturally occurring in humans and animals or monoclonal laboratory-produced—are considered excellent biorecognition elements because of their exceptional specificity and high affinity for their corresponding antigens. They have long been employed in biomedical applications, particularly in biosensors which use antibodies or ligands to influence antibody–antigen interactions. These biosensors are broadly categorized into labeled and non-labeled types. Enzyme labeling is a widely used bioanalytical method to attach a chemical marker to a molecule and it is applied in techniques such as ELISA, Western blotting, and immunostaining, where labeled antibodies are detected through a substrate reaction that emits light or induces a color change [65]. Antibodies can also be labeled with small molecules, radioactive isotopes, enzymatic proteins, or fluorescent dyes [37]. The idea of using immunological components as sensing agents was initially introduced in an immunoassay for measuring plasma insulin in humans [66]. The high dissociation constants (Kd) of antibodies for their target antigens have been widely utilized, with the ELISA test becoming the “gold standard” for immunoassays, serving as a benchmark for new immunoassays and immunosensors. ELISAs involve immobilizing a reactant (antibody or antigen) on a solid surface and using enzymes as markers to detect specific antibody–antigen interactions [67].

This technique, harnessing the high specificity of the Ab/Ag interaction, led to the development of the first commercially available immunoassay, the home pregnancy test for detecting human Chorionic Gonadotropin (hCG). Initially described in the 1960s, hCG immunoassays lacked specificity until the 1970s, when the first lateral-flow immunoassay was created to differentiate hCG from luteinizing hormone (LH). In the late 1980s, home pregnancy tests became available, and since then, the technology has expanded to various applications. By definition, the immunosensors integrate the transduction process to link specific Ab/Ag interactions to signal generation [66]. Various methods are available to convert the signal generated by the binding interaction between an antibody and an analyte, each with distinct advantages and disadvantages. These differences have fueled extensive research into the development of functional immunosensors and immunoassays. Contemporary research is particularly focused on the development of Point-of-Care (PoC) diagnostic systems, emphasizing the creation of portable, reusable and miniaturized devices. In parallel, significant efforts are being directed toward enhancing platform reliability and integrating innovative components such as aptamers, molecularly imprinted polymers, nanoparticles, and other advanced materials [67].

ELISA can be either direct or indirect, and there exists the sandwich ELISA and the competitive ELISA as well. The direct ELISA method is ideal for measuring high molecular weight antigens. In this technique, the plate surface is directly coated with either the antibody or antigen, and an enzyme-labeled antibody or antigen facilitates detection. After incubation, unbound antibodies or antigens are removed through washing. A suitable substrate is then added to generate a colorimetric signal, which is measured to quantify the antigen or antibody [68]. In the indirect ELISA method, diseased serum is added to antigen-coated wells, forming antigen–antibody complexes during incubation. A secondary enzyme-labeled antibody is then added to detect the complexes. Upon adding the enzyme’s substrate, a color change occurs, allowing for concentration measurement. This method is frequently used in endocrinology for antigen detection [69]. In the Sandwich ELISA method, a sample is added to antibody-coated wells and incubated. Washing removes unbound antigens, while specific antigens remain bound. Enzyme-labeled antibodies are then added, bind to these antigens, and are retained after further washing. Adding the enzyme’s substrate produces a color change, indicating a positive result, while no coloration indicates a negative result. This method, where the target protein is “sandwiched” between two antibodies, is 2–5 times more sensitive than other ELISA techniques [70]. In competitive ELISA, wells are coated with an antigen-specific antibody or antibody-specific antigen. The test sample and enzyme-tagged antigen or antibody are added simultaneously, competing to bind with the antibody or antigen on the well surface. After washing, substrate is added, and the resulting color intensity is inversely proportional to the analyte concentration. Low analyte levels produce high absorbance, while higher analyte amounts result in lower absorbance [71].

Nowadays, immunoassays have seen tremendous advancements, and the so-called Point-of-Care (POC) devices have been widely used—especially during the pandemic of COVID-19 [5,12]. POC Lateral Flow Immuno Assays (LFIAs) also referred to as immunochromatographic tests, are simple and rapid diagnostic tools designed to detect the presence of a target analyte—such as a specific antigen, antibody, or nucleic acid—in clinical specimens. These assays are widely employed in medical diagnostics due to their user-friendliness, portability, and ability to deliver results within a few minutes. This transformation aims to enhance decision-making efficiency and reduce turnaround times [72]. The test strip comprises several components: a sample pad, a conjugate pad containing labeled particles, a nitrocellulose membrane with both test and control lines, and an absorbent pad. As the sample flows through the strip, target analytes bind to the labeled particles, forming complexes that are subsequently captured at the test line, resulting in a visible signal. A control line is included to ensure the validity of the test [73]. LFIAs are widely used for their simplicity, speed, and portability but may have limitations. Immunobiosensors have various applications in clinical diagnosis and monitoring, as also seen in COVID-19 pandemic and further medical conditions. Such biosensors can mostly detect antigens and antibodies and can have various detection systems, such as electrochemical, optical and other methods, which will be discussed in further subsections [74].

Monoclonal antibodies in immunosensors are also capable of detecting whole cells, such as bacteria. Since enzyme-based, antibody-based, and other biosensing systems typically require additional reactions—therefore necessitating the use of reagents—biosensors designed for the immediate, reagentless detection of whole cells are highly desirable. When it comes to whole bacteria, impedimetric and optical methods are among the most commonly employed approaches. A variety of surface antigens on bacterial cell envelopes, including proteins, glycoproteins, lipopolysaccharides, and peptidoglycan, can serve as targets for biorecognition. However, bacteria, and cells in general, exhibit diverse surface epitopes that may provoke non-specific interactions with the sensor surface, particularly with the antibody [75].

Nevertheless, antibody (and antigen) detection is prone to false results and false positives in particular, since antibody detectors (like in ELISA) target mainly the sensitivity rather than the specificity [14]. Interference is particularly evident in blood specimens under certain conditions, including hemolysis, icteria and lipemia [76]. High levels of drugs and their metabolites—resulting from daily moderate-to-high consumption or intake shortly before sampling—can also cause interference and false results. Similarly, the consumption of certain foods and their metabolites (alcohol consumption particularly for LFIAs), the so-called Hook effect—that can trigger false negatives in high analyte concentrations-, as well as the presence of radioactive or fluorescent compounds that can result in false positives [76]. Biotin in particular is common for its negative inhibitory impact on antibody–antigen test results. Particularly, antibody-related interferences parameters that can trigger false results (mostly false positives) include the ability of the organism to produce antibodies (e.g., immunocompromised can yield false negatives) and genetic parameters (SNPs, mutations, no antibody gene expression, aggregated complexes, different post-translational modifications that may result in false negatives), and therapies with monoclonal antibodies resulting in high blood concentrations [76]. A major cause of immunointerference is the presence of heterophile antibodies in the tested clinical specimens. These include the Human Anti-Animal Antibodies (HAAAs) present in cases who have frequent interactions with animals/animal products. Other sources of interference include antibodies against the sensor antibody (i.e., anti-streptabidin/anti-ruthenium/anti-biotin antibodies), as well as antibodies that bind to or against the target analyte, leading to aggregations that cannot be detected from the sensor. Additionally, autoimmune-derived antibodies, the potential high levels of IgG4, which are commonly present in all people samples in low concentrations, and the circulating antibodies due to external factors—such as blood transfusion, transplants, maternal transfer, pregnancy and multiparous women, surgery and vaccinations—can also contribute to immunointerference. Except for heterophile antibodies, the presence of the so-called paraproteins in a clinical sample has the same impact; hypergammaglobulinemia, pseudohypophosphatemia, dyscrasia, multiple myeloma, lymphoma, Waldenstrom macroglobulinemia, amyloidosis, monoclonal gammopathy, light chain disease, cryoglobulinemia, and other conditions are characterized by high levels of paraproteins [76]. Generally, like in all tests performed in vitro in clinical specimens sampling/handling contaminations, sample/reagent degradation, temperature fluctuations, humidity, inadequate sample manipulations, poor calibration, issues with Load of Detection (LoD) score and relevant factors, as well as errors related to devices, can result in false results [5,12,14,76]. In addition to the issues discussed earlier in the enzyme-based sensors, one could argue that, aside from the interference and cross-reactivity, the stability of antibodies is critical. Over time, antibodies can lose their structural integrity over time, particularly under unfavorable storage conditions, leading to reduced binding affinity and sensor sensitivity. The method of antibody immobilization on the sensor surface is another key factor; improper immobilization can affect the orientation and accessibility of antibodies, diminishing their ability to bind effectively with the target analyte. Variability in antibody affinity can also cause inconsistent sensor responses, particularly if antibodies with lower affinity are used. Environmental factors, such as pH, temperature, and ionic strength, can influence antibody–antigen binding efficiency, further impacting sensor performance. Moreover, LoD score, sensor surface fouling may occur over time as antibodies undergo conformational changes or aggregation, leading to decreased sensitivity and efficiency. Finally, the cost and availability of high-quality, specific antibodies can limit the development of certain immunosensors, posing challenges in creating effective and affordable detection systems. The affinity and specificity of the antibodies play a crucial role in the sensitivity of immunosensors. Improper surface chemistry or antibody conformation during immobilization can compromise the sensor’s performance, while dilution and incubation time affect the quality of antigen binding. Additionally, temperature sensitivity or electrostatic interactions may lead to undesirable changes in antibody–antigen interaction, necessitating precise optimization of the isotype and buffer composition for optimal results.

#### 3.1.3. Whole-Cell-Based Biosensors

Microbial cell-based biosensors emerged in the late 1970s, utilizing cellular processes such as fermentation, metabolite production, or respiratory changes, combined with electrochemical transducers to measure these activities [77]. While initially cost-effective and innovative, they lacked the selectivity of enzyme-based biosensors that will be discussed in the following subsection, which dominated the field [77]. However, advancements in recombinant DNA technologies led to significant improvements and evolution in microbial cell-based biosensors. The use of whole cells in sensing systems, offers unique advantages, especially since only living cells can provide functional information—insight into how a stimulus impacts a living system—unlike analytical methods, which measure substance quantities. These functional data are crucial in situations where understanding the biological effects of a substance is the primary goal. In such cases, employing living cells becomes highly valuable as it directly delivers the required functional insights. Early examples of such scenarios include [78]:

(i) The potential effect of the specific drug on a given biological system and its possible role as either an agonist or and antagonist,

(ii) The potential secondary actions of this specific drug as on other aspects of the cellular metabolism,

(iii) The potential toxicity of the specific drug, and

(iv) The possible pollution of the environmental sample.

Whole-cell-based biosensors incorporate microorganisms such as fungi, bacteria, viruses, and algae due to their potential biological components. These microorganisms can replicate and produce recognition elements like antibodies without requiring extraction or purification. In this setup, cells can interact with a wide range of analytes, producing an electrochemical response that is easily detected and processed by a transducer [37]. Essential information can be obtained from the cells through various ways, including changes in pH, measurements of O_2_ consumption and CO_2_ production, estimation of the redox potential, measurement of the electrical potential on nervous system cells, or the production of certain metabolites [78]. Whole-based biosensors can offer their simplicity in conducting field tests, their ability to readily detect fractions containing bioavailable contaminants, their ability to measure the total bioavailability of a pollutant, rather than just its free form, as well as the continuous monitoring of substances. Initially, such biosensors were designed for environmental usage, since cells can act like as sensors against environmental analytes, and [79]:

(i) Traditional analytical methods based on physical and chemical techniques are often highly precise and sensitive in identifying the specific composition and concentration of toxins in samples; however, they are typically limited in assessing bioavailability, toxicity, and genotoxicity for only a small range of toxins, and in many cases, measuring critical parameters is only achievable through the use of living cells.

(ii) The performance of whole-cell biosensors for pollutant detection relies on reporter genes regulating transcription and their associated regulatory proteins, whereas these genes convert biological responses into detectable signals, crucial for the biosensor’s sensitivity and specificity.

(iii) The regulatory protein, which interacts intricately with the target analytes of the contaminants, plays a vital role in ensuring their specificity and sensitivity.

(iv) The choice of host cell significantly affects a biosensor’s specificity, sensitivity, and response time, while due to their metabolic and genetic similarity to host organisms, eukaryotic cells are used in most of the biosensors for metal detection.

(v) While whole-cell biosensors generally outperform traditional chemical-based sensors, ongoing research focuses on enhancing their accuracy, and practicality; this effort maybe stems from growing concerns over escalating pollution levels.

Living-cell-based biosensors have gained significant interest among researchers, driven by advancements in microfabrication and cell immobilization techniques. These biosensors can measure quantitative data about the condition of live animal or bacterial cells by translating homeostatic signals into detectable electrical or optical outputs. Cells naturally possess molecular recognition components like receptors, ion channels, and enzymes, which are highly responsive to specific analytes due to their inherent cellular mechanisms [80]. Whole-cell-based biosensors leverage this sensitivity to monitor and analyze various physiological parameters, such as metabolic changes, impedance, and action potential, under external stimuli. They provide insights into biological metabolic disorders and other pathological conditions at a cellular level, making them highly valuable in biomedicine. Their applications extend to areas such as cellular physiology studies, pharmaceutical testing, and medical diagnostics.

A key goal of current precision medicine research is to investigate genetic mutations in drug targets, such as G Protein-Coupled Receptors (GPCRs), and how these mutations alter their responses to various drugs [81]. Whole-cell-based biosensors have also been used for the detection of micronutrients, as well as for the identification of pathogens [82,83].

It has been discussed that the distinction between Microbial Surface-Display Biosensors (MSDBs) and Microbial Whole-Cell Biosensors (MWCBs) is crucial; while both utilize microbes in their design, their sensing mechanisms and functionalities differ significantly, categorizing them as distinct types of biosensors [84]:

(i) In MSDBs, microbes are genetically modified to express foreign proteins on their cell surfaces by fusing them to anchoring motifs of outer membrane proteins, and the activity of these surface-displayed proteins, such as their inhibition or catalytic response to target analytes, is measured to quantify the analyte levels; in that case, the surface-displayed enzyme acts as the primary sensing element, closely linked to a transducer, while the microbes function as a supporting matrix for immobilization.

(ii) In MWCBs, the analyte recognition is integrated with signal-generating proteins to create sensing microbes which contain modified DNA with sensing and reporter genes, produce measurable signals in response to target analytes, an unlike MSDBs, where microbes act as matrices, MWCBs use entire cells as sensing elements and their design enables selective, dose-dependent detection of bioavailable analytes by leveraging precise protein–ligand interactions.

Various MWCBs have been developed and utilized to monitor environmental and agricultural systems, assess micronutrients, evaluate metallic trace elements and organic compounds as well as detect quorum sensing molecules. They are also widely employed in biomedical applications with most relying on fluorescent, bioluminescent, or electrochemical reporter proteins [84]. Biomedical diagnostic techniques, which utilize whole-cell biosensors, demonstrate significant promise and potential within the field of biomedical engineering.

Regarding misdiagnoses, several factors can trigger false results in these types of biosensors. As with previous conditions, interference remains a concern, as cells may interact with unintended substances, causing false positives if the biosensor lacks sufficient specificity. Additionally, background noise—such as the presence of other chemicals, toxins, or contaminants—can interfere with the biosensor’s signal, leading to either false positives or false negatives. In addition to fluctuations in pH, temperature, and other environmental conditions, cells may adapt to the detection environment over time, which can reduce their responsiveness and affect the accuracy of subsequent readings. Moreover, incorrect pairing can result in false positives, and genetically modified strains may evade proper probe matching. However, this process remains time-consuming and costly [75]. Actually, genetically modified or engineered cells can undergo mutations that affect the accuracy of their response or can generate mutated products, leading to potential false results. Improper or suboptimal cell immobilization may result in loss of functionality or hinder the sensor’s ability to interact with analytes, while cells may adapt to the detection environment, causing them to lose their responsiveness over time and affecting the accuracy of subsequent readings [79,85]. Furthermore, overexposure to the target analyte or prolonged use can lead to desensitization of cellular receptors, affecting the biosensor’s ability to detect analytes accurately. Additionally, alterations in cellular metabolism due to external stimuli or the presence of the target analyte may impact the sensor’s output and result in inaccurate readings. There is also a potential impact on cellular vitality [86,87]. Also, high concentrations of analytes could overload the sensing system, leading to a limited or skewed response that does not reflect true analyte concentration. If the receptors on the cell surfaces are saturated with analytes or other molecules, it may cause misleading results due to reduced sensitivity. Damage to the cell membrane, whether from harsh handling or prolonged exposure to test substances, can impair the cell’s ability to function as a proper biosensor [88]. Inaccurate results in whole-cell biosensors can also arise from issues like inconsistent cell growth, inactivation of reporter genes, improper cell immobilization, low analyte concentrations, inadequate nutrient supply, delayed response time, cellular degradation over time, and alterations in cell pathways [79].

#### 3.1.4. Nucleic Acid-Based Biosensors

Nucleic acid-based biosensors have garnered increasing attention due to their sequence variability and specificity, from single-nucleotide polymorphisms and methylation patterns to complementary strand pairing and unique secondary structures, enabling highly specific detection of target genetic biomarkers or disease-related alterations [89]. These biosensors, which can be DNA, RNA or aptamer based, typically utilize single-stranded DNA (ssDNA), which can hybridize with its complementary strand, offering exceptional efficiency and precision, thus making the detection of complementary DNA or RNA strands straightforward [89]. Moreover, such sensors enable the detection of various other analytes by employing functional nucleic acids such as aptamers or DNAzymes, as probe molecules. This extends beyond the traditional roles of nucleic acids, demonstrating versatile functionality in biosensing applications. To date, there exist various examples of DNA sensors for the detection of divergent targets such as viruses and cancers. The development of functional nucleic acids has expanded the capabilities of modern functional nucleic acid biosensors. These sensors now go beyond traditional gene detection to include the detection of metal ions, small molecules, proteins, bacteria, and more [89].

Electrochemical functional nucleic acid biosensors utilize functional nucleic acids integrated with an electrochemical transducer to transform the interaction of analytes with the electrode into detectable electrical signals. This sensing approach offers advantages such as reusability, high stability, and strong affinity, making it highly effective for detecting ions, enzymes, proteins, viruses, and cells. Fluorescent biosensors consist of three main components: a recognition element, a transformation mechanism, and a fluorescent material. Due to the ability of functional nucleic acids to be modified with fluorescent tags and quenchers, along with their strong recognition capabilities, these biosensors are widely used in various applications. They are generally categorized into two types: labeled fluorescent FNA biosensors and label-free fluorescent FNA biosensors [90]. Nucleic acid sensors demonstrate excellent selectivity due to the high affinity and structural versatility of NA molecules, while their sensitivity largely depends on the efficiency of the signal transducer and amplification systems. Generally, a variety of signal transduction methods are utilized in functional nucleic acid sensors, including fluorescence, electrochemistry, electrochemiluminescence, chemiluminescence, colorimetry, and SPR. To enhance sensitivity, several amplification techniques have been integrated into these sensors, focusing on amplifying either the target molecule or the detection signal.

The specific hybridization capability between nucleic acid strands is a fundamental principle of DNA sensor detection. This straightforward approach is often preferred in diagnostic laboratories over direct sequencing methods. The sensitivity of these measurements depends on several factors, including the methods used for sequence amplification—such as Polymerase Chain Reaction (PCR)—the effectiveness of sequence hybridization, and the degree of background signal interference. Particularly, PCR amplifies specific DNA sequences via utilizing cycles of denaturation, primer annealing, and extension with a thermostable DNA polymerase, enabling exponential replication of the target DNA region [91].

Through techniques like in vitro selection or Systematic Evolution of Ligands by Exponential Enrichment (SELEX), it becomes possible to evolve nucleic acids in laboratories to bind a wide variety of analytes beyond just DNA or RNA, demonstrating remarkable affinity and specificity. These binding nucleic acids are referred to as “aptamers” (the term “aptamer” originates from the Latin word “aptus”, meaning “to fit”, reflecting the lock-and-key relationship between aptamers and their target molecules) [92]. Aptasensors offer numerous advantages, such as strong affinity, straightforward preparation, and the ability to form Watson–Crick base pairs, making them capable of targeting a wide range of substances. These include inorganic ions, organic molecules, large biomolecules like peptides and proteins, and even entire organisms like bacteria and cells [89].

DNAzymes, a type of catalytic nucleic acids, can cleave specific substrates in the presence of cofactors such as metal ions or amino acids like L-histidine. These DNAzymes typically comprise a substrate strand and an enzyme strand. The substrate strand contains a single RNA linkage that acts as a cleavage site, while the enzyme strand includes a catalytic core and two binding arms. When a cofactor is present, the enzyme strand cleaves the substrate strand into two fragments, enabling the development of cofactor-responsive biosensors. Another type of DNAzyme, known as the G-quadruplex DNAzyme, undergoes structural transformation into a parallel or antiparallel G-quadruplex configuration in the presence of certain metal ions. This G-quadruplex DNAzyme exhibits peroxidase-like activity when combined with hemin, making it useful as a recognition element in biosensors or as a versatile label for various targets through rational design [93]. DNAzymes are highly versatile functional nucleic acids capable of integrating with fluorescent tags, functional molecules, and solid surfaces, making them ideal for diverse applications in biosensors. They are often designed as mediators in these systems, leveraging their unique properties for detecting and analyzing a wide array of substances. DNAzymes can form catalytic beacons that generate vivid fluorescent signals by attaching fluorescent groups and quenchers, a process reliant on specific metal ions as cofactors [90]. Additionally, DNAzymes bind with hemin to form robust G-quadruplex structures that mimic peroxidase-like activity, enabling precise signal transduction through their connection with affinity ligands. Their versatility has led to innovative applications in environmental monitoring, food safety detection, and sensitive liquid analysis, solidifying their role as indispensable tools in biosensing [90].

In addition to DNAzyme, deoxyribonuclease I (DNase I) has also been utilized as a mediator in functional nucleic acid biosensors. DNase I is an enzyme capable of hydrolyzing both ssDNA and double-stranded DNA (dsDNA). Although DNase I has traditionally been avoided in conventional sensors due to its ability to break down nucleic acid chains, advancements in functional nucleic acid biosensor research have expanded its application as a mediator. Researchers have discovered that DNase I can enhance signal strength by degrading these biosensors. This degradation releases the analyte, which then interacts with probes and other recognition elements, enabling repeated signals for effective signal amplification [94]. DNase I and DNAzyme are often used in these biosensors with the assistance of cofactors to construct mediators, as their activity depends on specific sequences. However, their reliance on cofactors presents challenges, particularly in signal amplification strategies, which limits their broader use as mediators.

Exonuclease III (Exo III) is a sequence-independent enzyme, meaning it does not require specific recognition sites for its application. Initially used in DNA sequencing due to its preference for cleaving dsDNA with fewer than four single nucleotides at the 3′ end, Exo III has become a key focus in FNA biosensor research [95]. With advancements in its study, Exo III-assisted cyclic amplification has emerged as a powerful signal amplification strategy for functional nucleic acid biosensors, enabling accurate and rapid detection of target substances. Moreover, Terminal deoxynucleotidyl Τransferase (TdT) is a DNA polymerase that adds deoxynucleotides to the 3′ hydroxyl end of DNA or RNA molecules. Like Exo III, it is sequence-independent and can use dsDNA or ssDNA with various end structures as primers. Its highest catalytic efficiency is observed with a 3′ protruding end. Consequently, researchers are focusing on methods to generate the 3′-OH end, either through restriction enzyme digestion or sequence design, to expand TdT’s use as a mediator in these biosensors [96]. Of course, up to now, more enzymes are used in this type of biosensors. Ιn addition, metal-based mediators in functional nucleic acid based biosensors offer high sensitivity, flexibility, and a broad detection range, making them valuable in environmental monitoring, food safety, molecular biology, and medicine. Currently, the primary types of metal mediators include metal–organic frameworks, metal nanomaterials, and ionic metals [90].

False results are evident in nucleic acid biosensors for various reasons. PCR has been discussed to trigger false results due to contamination of samples with DNA from unintended sources can lead to false positives. Additionally, primer–dimer formation due to incorrect primer design or concentration may cause non-specific amplification. Non-specific binding of primers to unintended sequences can also result in false positives [12,14]. In addition, poor quality or degraded DNA templates can lead to inconsistent or absent amplification, and improper reaction conditions—such as incorrect temperatures, pH, or salt concentrations—can negatively affect enzyme (polymerase) efficiency and cause false results [5,12,14]. Inaccurate DNA quantification, either too much or too little template, can lead to amplification issues, while cross-contamination between samples during handling can lead to false results, and the presence of either endogenous or exogenous inhibitors, can interfere with the PCR reaction, leading to false negatives. Furthermore, aging or improperly stored reagents may lose effectiveness, impacting PCR accuracy, and malfunctioning PCR machines, including incorrect heating or cooling, can prevent proper amplification or cause inconsistent results [5,12,14]. In a similar way, non-specific binding occurs when probes or aptamers bind to unintended targets, leading to false positives in nucleic acid-based biosensors. Also, incomplete hybridization, where complementary strands fail to bind correctly, fully or strongly or due to temperature variations, can also result in inaccurate measurements [97]. Contaminants or impurities in the sample can interfere with signal detection, and secondary structures formed by DNA or RNA sequences may obstruct proper binding or detection. Temperature fluctuations during testing can affect hybridization efficiency or enzyme activity, while inadequate probe concentration, whether too high or too low, can lead to inconsistent results. Enzyme inhibition or degradation may cause weak signals if the detection enzymes are compromised. Further, inaccurate calibration of the sensor can result in erroneous results, and cross-reactivity with other nucleic acids or similar sequences in the sample can generate false results, and also probe degradation over time can prevent proper binding to targets, leading to false negatives [97,98].

Aptamer-based nucleic acid biosensors can yield false results due to non-specific binding that is a common issue, where aptamers may bind to unintended targets or molecules that resemble the target, leading to false positives. Structural instability is another challenge, as aptamers rely on a specific three-dimensional structure to bind effectively. Any alteration in this structure—due to environmental conditions such as temperature or pH—can disrupt proper binding, resulting in false negatives [99,100]. Incomplete binding, caused by steric hindrance or competition from other molecules, can also lead to weak or inaccurate signals. Moreover, interference from the sample matrix, including proteins or ions, can affect the aptamer’s ability to bind its target, leading to unreliable results, and additionally, degradation of the aptamer over time and the use of inappropriate aptamer concentrations can further exacerbate false results [99,101]. DNAzyme-based nucleic acid biosensors face similar challenges including the inhibition or degradation of DNAzyme activity, which may occur due to environmental factors such as the presence of inhibitors or contaminants, causing false negatives. Substrate interference is another concern, where non-specific substrates or molecules can compete with the target substrate, disrupting DNAzyme’s catalytic activity and leading to incorrect measurements [102]. DNAzymes also require specific metal ions or cofactors for optimal function, and any discrepancies in the cofactor concentration can hinder the catalytic process, resulting in false results. Also, incomplete reactions due to time or temperature constraints can lead to inaccurate signal generation. Lastly, the formation of secondary structures within DNAzymes can prevent proper interaction with their target substrates, leading to a lack of catalytic activity and false-negative results [90,93]. Of course, one must bear in mind that again pH stability, sensitivity to nuclease degradation and the costs of the aptamer synthesis can give rise to false results in aptamers.

Figure 1 illustrates reasons for false results in biosensors due to bioreceptors.

### 3.2. Current Common Biosensors Based on Their Transducers-Detection System

#### 3.2.1. Electrochemical Biosensors

Since the electrode’s discovery in 1950s, tremendous advancements have been achieved in biosensors for more than a half century. Biosensors are created to detect and monitor biological reactions occurring on the surface of transducers. A crucial requirement in developing an electrochemical biosensor is the consistent immobilization of target biomolecules onto the sensor surface while preserving their biological activity. Electrochemical detection is widely used in biosensors due to its low cost, simplicity, portability, and ease of construction. Electrochemical detection typically measures current (amperometry), charge or potential changes (potentiometry), or variations in conductivity (conductometry), whereas electrochemical impedance spectroscopy, which assesses both resistance and reactance, is increasingly being adopted for enhanced biosensor performance. Electrochemical biosensor parts include a biorecognition element immobilized on a conductive surface, a transducer which converts biological interactions into electrical signals, and a signal processing unit for data analysis.

Electrochemical detection in biosensors offers key advantages, including minimal dependence on reaction volume and the ability to analyze very small samples, whereas also it enables ultra-low detection limits in immunoassays with little to no sample preparation, achieving high sensitivity [103]. Unlike spectrophotometric methods, electrochemical detection remains unaffected by interfering sample components like chromophores, fluorophores, or particulates, making it ideal for analyzing colored or turbid samples such as whole blood. Electrochemical techniques are categorized into current, potential, and impedance measurements, with current-based methods being the most widely used in biosensors [103]. Electrochemical biosensors can indeed be enzyme, immuno, whole cell, and nucleic acid based, whereas all these types utilize electrochemical transducers to convert biological interactions into measurable electrical signals. Here, we will discuss these common electrochemical biosensors.

##### 3.2.1.1. Voltametric/Amperometric Biosensors

Voltametric and amperometric techniques involve applying a potential to a working (indicator) electrode relative to a reference electrode while measuring the resulting current. This current arises from electrochemical oxidation or reduction at the working electrode and is influenced by the rate at which molecules are transported to the electrode surface [104,105]. Voltammetry specifically refers to methods where the applied potential is systematically varied over a defined range, generating a current response that appears as a peak or plateau, directly proportional to the analyte concentration. Various voltametric techniques exist, including linear sweep, cyclic, hydrodynamic, differential pulse, square-wave, AC voltammetry, polarography, and stripping voltammetry; these methods offer a broad dynamic range and high sensitivity, making them particularly effective for detecting and quantifying low analyte concentrations. Voltametric biosensors use various techniques to measure electrochemical signals, with square-wave voltammetry providing high sensitivity and resolution via a series of square-wave pulses; cyclic voltammetry providing insights into the redox behavior of analytes via a repetitive potential scan; and differential pulse voltammetry enhancing sensitivity through measuring the current difference between two adjacent potential pulses [104]. Voltammetry involves measuring the current as a function of varying the potential (voltage) applied to the working electrode. This method provides information about the electrochemical properties of the analyte, such as its redox behavior, and the current changes are observed while the potential is scanned over a range.

On the other hand, amperometry measures the current at a constant potential applied to the working electrode. The current is directly proportional to the concentration of the electroactive species in the sample, and no potential scanning occurs during the measurement. It was discussed in previous paragraph that amperometric biosensors were first introduced in 1962 by Leland Clark, who introduced an amperometric enzyme electrode for glucose detection [25]. Amperometric biosensors were the first to be created and have been used as glucose sensors for more than 35 years. An amperometric biosensor is a device that measures the current between electrodes during a redox reaction, commonly used to detect and quantify glucose levels in a sample. Amperometry measures changes in current caused by the electrochemical oxidation or reduction of analytes at a constant potential [105]. Unlike voltammetry, no scanning potential is used, and the potential is stepped to a specific value, and current is measured, which correlates to the concentration of the electroactive species. Amperometric biosensors are a preferred technique for all these reasons and also because they offer enhanced selectivity by using unique oxidation/reduction potentials for different analytes [105].

Electrochemical cells can have either two or three electrodes: (i) a working electrode (gold, carbon, or platinum) for analyte detection, (ii) a reference electrode (Ag/AgCl) to maintain a stable potential, and (iii) a counter electrode to complete the circuit and support current flow. The reference electrode remains distant from the reaction site to maintain a stable potential, while the auxiliary electrode handles the charge during electrolysis [35,106]. A two-electrode system only has a working and reference electrode, and can handle the charge if the current density is low, and it is often used for disposable sensors due to lower cost and less need for reference stability. Small electrode sizes, including those in the micrometer and nanometer range, improve sensitivity and are suited for small sample volumes [35,103,107]. Additionally, these miniaturized systems can be manufactured affordably for portable biosensing instruments, though maintaining a constant temperature is important for accurate results in voltammetry. Screen-printed electrodes are low-cost, easy to mass-produce, and ideal for miniaturization, making them popular in electrochemical biosensors and portable systems and they are commonly used for detecting trace metals, pesticides, and in glucose sensors [108,109]. Interdigitated array electrodes feature interwoven metal strips that enhance redox cycling, improving detection sensitivity by increasing the signal-to-noise ratio. They are used in immunoassays and can achieve significant signal enhancements depending on the electrode dimensions [110]. Amperometric biosensors are classified into three generations based on their electronic transport mechanisms [105]:

(i) First-generation biosensors measure the concentration of electroactive enzymatic reaction products, typically using oxidases and dehydrogenases to detect oxygen reduction or hydrogen peroxide generation;

(ii) Second-generation biosensors introduce redox mediators, like ferrocene, to shuttle electrons between the enzyme and the electrode, enabling detection even for enzymes with electrically insulated active sites;

(iii) Third-generation biosensors bypass mediators by achieving direct electron transfer between the enzyme and electrode, enabled by advanced immobilization techniques and surface modifications to optimize enzyme orientation and electron exchange.

Despite their various advantages, including the minimal dependence on reaction volume, the ability to analyze very small samples, the good linear ranges and the low LoD, the precision of voltametric analysis is often constrained by the need to account for residual currents, especially those caused by charging effects. At analyte concentrations in the parts-per-million range, an accuracy of ±1–3% is typically achievable, yet, as the analyte concentration decreases significantly, the accuracy tends to decline [111]. It has been discussed that in protein voltametric analysis, intermediary layers (or interface layers) which are materials placed between the electrode surface and the analyte solution to enhance sensitivity, selectivity, and stability, play a crucial role in facilitating electron transfer, minimizing interference, and improving protein immobilization. However, these layers have several limitations. For example, self-assembled monolayers in protein voltammetry may lead to false results due to time-consuming preparation, poor conductivity limiting electron transfer, defect formation, easy desorption during redox processes, and low wettability and corrosion resistance. On the other hand, carbon-based nanomaterials such as graphene, may cause false results due to their low dispersibility in water, which arises from hydrophobicity, and their tendency to agglomerate due to high surface area and surface energy. Similarly, carbon nanotubes are prone to erroneous results because of their nonbiodegradability, strong tendency to aggregate, lack of solubility in aqueous solvents, and challenges in producing structurally and chemically reproducible batches with minimal impurities [112]. Metal/metal oxide nanoparticles can cause false results due to their tendency to agglomerate, susceptibility to surface oxidation and corrosion, and their potential to alter protein stability, structure, and function, leading to denaturation. Quantum dots have low stability and higher power consumption. Similarly, magnetic nanoparticles face challenges such as poor dispersion in aqueous solutions and chemical instability [112].

False results in the voltametric analysis of peptides can arise due to several factors. The requirement for a high negative potential limits electrode selection, while sensitivity variations in catalytic hydrogen evolution may interfere with measurements. In addition, donor chemical species can affect proton transfer, and reduced transition metal complexes can introduce signal variability. Limited electroactive amino acids in the anodic region may cause false negatives. The irreversible and multistep nature of amino acid oxidation reduces precision. Finally, reliance on peak position, height, and shape, which can be influenced by instrument settings, sample preparation, or matrix effects [112]. In addition to the non-specific redox reactions, residual currents, contaminants, secondary species, electrode fouling and instability, other factors such as redox cycling, instrumental drift, incomplete rinsing, cross-reactivity and electromagnetic interference from external noise, are also likely to lead to false positives in voltametric assays [35,113,114,115,116]. Therefore, except for the low presence of the analyte, the short deposition time and the variances in pH and ionic strength, poor mass transport, signal suppression by complex matrices, inefficient range, passivation of electrodes and competitive adsorption can trigger false-negative test results [35,103,117,118].

Amperometric false results are similar to those observed in voltametric assays and can arise due to interference of similar electroactive species that can be reduced/oxidized at the same potential as the target molecule. Additionally, non-specific adsorption can lead to spurious signals, while contaminants, external noise, matrix composition, electron fouling and instrumental drift can further contribute to inaccuracies, much like in voltammetry [37,119,120]. Also, if the electrode potential is kept at an overly high level for too long, it can result in unwanted side reactions that give false positive signals.

##### 3.2.1.2. Impedimetric Biosensors

Since 1975, various strategies have been utilized to design impedimetric biosensors, which identify biological interactions via measuring alterations in the electrical impedance at the interface of the sensor that is caused by biomolecular binding or surface alterations [45,121]. It was discussed in the previous subsection that electrochemical impedimetric biosensors measure electrode impedance, analyzed by EIS to detect biorecognition events. EIS analyzes both the resistive and capacitive properties of materials via applying a small-amplitude sinusoidal excitation signal to a system at equilibrium [122]. The impedance spectrum is obtained by varying the frequency over a broad range, allowing the measurement of in-phase and out-of-phase current responses to determine the resistive and capacitive components, respectively [103]. Impedance techniques are highly effective, as they enable the analysis of electron transfer at high frequencies and mass transfer at low frequencies. Due to time constraints and specific requirements, EIS is less common in enzyme-based biosensors and primarily serves as a characterization tool. However, impedimetric biosensors—an affinity-based type of biosensor—can be enzyme immuno, whole cell or nucleic acid based [45]. In impedimetric immunosensors, the electrode surface is often modified with a specific biological recognition element, which is typically incorporated into a conductive polymer film via electrochemical deposition. During detection, a known voltage is applied, and the resulting current is measured; when the analyte binds, the electron transfer resistance at the electrode-solution interface changes [103]. This label-free method offers advantages like higher sensitivity, lower costs, faster assays, and quicker response times, yet regenerating the sensor surface for repeated measurements is usually time-consuming and unreliable, and this is their highest challenge. Further, the regeneration process can also damage the immunoreagent attached to the transducer surface, causing it to be released [103].

Similarly, there are various reasons for which false results can arise in impedimetric biosensors. Erroneous impedimetric results can arise from variations in electrode pretreatments, which alter the physicochemical properties of the gold surface, subsequently impacting capacitance and topography. Inconsistent pretreatment effects on impedance measurements (like resistivity), poor repeatability of inter-electrode measurements, large standard deviations in sensor responses, and insufficient surface preparation can all contribute to false results in aptasensor impedimetric testing [123]. Incomplete blocking of the electron surface due to insufficient blocking agents, poor pretreatment and cleaning, fouling of the electron surface, cross-reactions, impurities and contaminants, as well as incorrect chemical coupling are also possible reasons of a false result in impedimetric biosensors [37,124,125]. Incorrect ligand orientation on the sensor surface can lead to misleading results in impedimetric assays. This may cause false positive interactions due to inaccessible or blocked binding sites, non-specific interactions, unstable and poor binding, inconsistent functionalization of the sensor surface, steric hindrance, aggregations and altered electron transfer properties [126]. Not only variations in temperature and pH, but also noise due to external electrons, power fluctuations and interference by external electronic devices can affect impedance readings or mask the usable signal [127,128]. Also, poor calibration of the sensor, unintended redox reactions on the electrode surface, variances in sensor fabrication, regeneration issues as well as leakages and the formation of liquid and mostly air bubbles are some major flaws in impedimetric assays [127,128,129,130].

##### 3.2.1.3. Potentiometric Biosensors

Since 1969, potentiometry—which is applied mostly for the identification of metal ions in clinical samples but also for the detection of enzymes, bacteria, DNA, aptamers, proteins and toxicity—has experienced a renaissance. This resurgence is driven by improved detection limits, selectivity, novel materials, advanced sensing concepts, and better theoretical modeling, driving innovations in ion sensing and biosensing. Polymeric membrane electrodes are a well-established analytical tool for electrolyte analysis, valued for their small size, rapid response, ease of use, low cost, and resistance to color and turbidity interference [131]. Potentiometric biosensors measure the potential difference across a selective membrane in an electrochemical cell with minimal current flow, using reference electrodes to detect specific ions. These biosensors are created by coating the membrane with a biological element (e.g., enzyme), which catalyzes a reaction producing the target ion for detection [131]. The most common types of potentiometric sensors include the membrane-based ion-selective electrodes (ISEs), screen-printed electrodes, ion-selective field effect transistors, solid-state devices, and chemically modified electrodes (using, e.g., metal oxides or electrodeposited polymers as sensitive layers) [132]. Also, some common biosensors are the [133]:

(i) LAPS; are semiconductor-based potentiometric sensors which utilize a light probe to manage the sensor response, enabling spatially resolved biological or chemical sensing;

(ii) Molecularly imprinted polymer-based potentiometric biosensors use molecularly imprinted polymers to create highly selective recognition sites, boosting the specificity and sensitivity of biosensing systems for accurate identification of the target analytes;

(iii) Wearable potentiometric sensors incorporate advanced materials and electronics into flexible and body-compatible designs, thus enabling real-time non-invasive control of important physiological biomarkers present in body fluids;

(iv) Photoelectrochemical potentiometric sensors leverage photoactive materials in order to enhance biomarker identification sensitivity via generating electrochemical outputs by light absorption, thus enabling selective and high-performance biosensing applications.

ISE biosensors are widely available for determining the pH of a clinical specimen, as well as the Na^+^, K^+^, Ca^2+^ and Cl^−^ concentrations, enzyme-based potentiometric biosensors are commonly use for the detection of urea, creatinine, cholesterol, MIP biosensors are used for identifying dopamine, histamine, cortisol and Prostate-Specific Antigen (PSA), and photoelectrochemical biosensors can be used for DNA hybridization and the detection of heavy metal ions and cancer biomarkers [131,132,133].

There exist various reasons that could trigger false results in potentiometric biosensors, such as fluctuations in pH, oxygen, temperature and humidity, electrostatic and electromagnetic interference [134,135]. Inhibition and interference are common issues in potentiometric biosensors due to the complex matrices of clinical samples. Cross-reactions with non-target analytes, such as ammonia and lipids, as well as sample contaminants like surfactants or detergents, can impact accuracy. Additionally, imbalances or unstable electrodes—especially the reference electrode—can lead to potential drift and erroneous results [131,136,137]. Interference in potentiometric biosensors is time and concentration dependent. Interference can also arise due to factors such as gas permeability, acidity, solubility, inward and outward flux species, alterations in internal filling electrolyte composition, equilibrium selectivity. Prolonged steady-state measurements, diffusion-layer effects in solid-state ISEs that trigger complex interference, membrane ion-exchange competition and modified diffusion processes can further complicate interference [137]. In addition, membrane degradation, loss of biocatalytic stability/inactivation as well as sample activators or inhibitors contribute to potential inaccuracies [137]. For inhibitions, it has been shown, for instance, that mercury and other metal ions have been found to inhibit potentiometric urea biosensor [138]. Similarly to other biosensors, erroneous results in potentiometric measurements can arise from improper calibration, sensors aging/degradation, improper storage conditions and manufacturing issues.

One could argue also that poor light source and intensity, erroneous light wavelength, unsuitable thickness of the insulating layer, low-sensitive photoactive material and light scattering may trigger misdiagnoses in LAPS. MIP-based potentiometric biosensors may yield misleading results due to poor/wrong polymer template selection and polymerization (and its conditions), failure in removing the template, poor binding interactions and polymer instability, cross-reactions and interferences and overloading (high analyte concentrations) as well, erroneous ratio of template to monomers and low analyte concentrations and matrix issues (i.e., viscosity) [139,140,141,142]. Photoelectrochemical potentiometric biosensors may lead to misdiagnoses due to several factors, such as light source malfunctions, improper light exposure (either too high or low), or external light interference, which can cause saturation and hinder detection [137,143]. Other contributing factors include degradation or inconsistent coverage of the photoactive material, poor charge carrier mobility, noise, low photoelectrode sensitivity, unstable or poor electrical contact, and delayed response times [137,143,144].

##### 3.2.1.4. Conductometric Biosensors

Conductometry is a method in which electrochemical reactions on the electrodes are minimal or absent, with the primary focus on the conductivity of the electrolytic solution, which alters in response to multifarious biological reactions [145]. Since 1961, conductometric detection detects variations in the electrical conductivity of a sample solution or a medium, as the concentrations in the solution or medium changes during a chemical reaction. Typically, these systems incorporate enzymes, whose charged products cause changes in ionic strength, leading to an increase in conductivity [103]. Conductometric biosensors function by detecting changes in the conductivity of a solution as ions are produced during a biochemical reaction. The movement of these ions, induced by an applied electrical field, generates a current that is measured to determine the concentration of the target substance. The transducer in that case is a miniature two-electrode device, which is created to estimate the conductivity of the thin electrolyte layer which is adjacent to the surface of the electrode. The enzyme reaction leads to a change in the conductivity of the solution near the electrode, and this change is detected by the conductometric transducer [146]. Various types of conductometric biosensors, including those based on direct analysis (primarily for biomedical applications such as detecting proteins, urea, arginine, and non-metal ions), inhibition analysis, whole-cell and DNA biosensors, as well as those for identifying microorganisms, have been extensively discussed, with applications spanning both environmental and agricultural industries [145].

False results in conductometric biosensors are not uncommon, and among their causes are the ingredients’ wrong ratio or absence of active substances at all [147]. Therefore, as with other types of biosensors, potentiometric biosensors are also susceptible to potential false results due to fluctuations in temperature, pH, and ionic strength, as well as chemical reactions, ion interferences, sensor drift, contamination, electrode fouling, reduced enzymatic activity, low sample volume, electrochemical noise, overload, and poor membrane/coating quality.

#### 3.2.2. Optical Biosensors

One of the earliest optical biosensors for clinical use was a glucose test strip for urine, which was commercialized in 1957 [46]. This sensor employed a cellulose pad with co-immobilized glucose oxidase and peroxidase to catalyze glucose oxidation, producing a visually detectable blue color for semi-quantitative urine glucose measurement [46]. Optical transducers detect changes in optical properties, such as fluorescence intensity, light absorption, reflectance, chemiluminescence, evanescent wave, refractive index, and Raman scattering, resulting from the interaction between a biocatalyst and the target analyte. Immunosensors were presented in previous subsections and SPR will be discussed in the next subsection. Here, we will discuss some common optical biosensors.

##### 3.2.2.1. Colorimetric Biosensors

Colorimetric biosensors work through detecting alterations in the color of a substrate, which takes place when a biochemical reaction, like the enzyme activity or antigen-antibody binding, leads to a chemical alteration which changes the light absorbance or reflection of the material, thus making it visible for the naked eye [148]. A colorimetric sensor identifies an analyte by exploiting the optical characteristics of the sensor, such as absorption or reflection, to produce a structural color that acts as the output signal for detecting the analyte. Monitoring the refractive index of an analyte by observing the color shift on the sensor’s surface offers several advantages, including ease of use, low cost, compatibility with on-site analysis, and real-time detection [148]. Colorimetric sensors have gained significant interest due to their speed, simplicity, high sensitivity, and selectivity. The signal can be detected via nanoparticles (such as golden nanoparticles (AuNPs) in LFIAs), peptides and antibodies (i.e., in ELISA), MIPs, aptamers, enzyme-assisted, whole cells and others (some were discussed in previous subsection) [148]. The sensing mechanisms of the colorimetric sensors are commonly based on nanoparticle aggregation, nanoparticle decomposition, nanozymes, fluorescence on–off, ligand–receptor interactions, and photonic structures [149]. For instance, the Loop-mediated Isothermal Amplification (LAMP) technique is a highly sensitive DNA amplification method that can detect specific genetic sequences with rapid color changes, making it ideal for colorimetric cancer detection and other molecular diagnostics [150].

There exist various reasons for a false result in colorimetric biosensors. Firstly, one could argue that because of the subjectivity of color perception, misdiagnoses may occur, changes in the surrounding light and the background color interference can modify the perceived color, while photobleaching and photochemical destruction can also trigger false results [151,152]. Moreover, the recent COVID-19 pandemic made it easier for LFIAs to be studied on the context of false test results, like interference and inhibitors [5,7,8,12,13]. Variations in humidity, temperature, turbidity, viscosity, ionic strength and pH, as well as oxidation effects that can alter chromophore states and induce color changes, can cause interferences with colored compounds [5,7,8,12,13]. Additional factors such as inconsistent light source, light-scattering nanoparticles, autofluorescence from microfluidic chips in polymeric devices or fluorescence due to food/drug products and precipitation of indicators can all lead to false results [5,7,8,12,13,150,152,153]. Other reasons include inhomogeneous reactions and ununiformed dying and color distribution (commonly due to inhibition or low analyte), chromatic aberration, aggregated nanoparticles resulting into artifacts and chemical contamination due to endogenous or exogenous factors [151,153,154,155,156].

##### 3.2.2.2. Fluorescent Biosensors

The Nobel-winning development of Aequorea Victoria Green Fluorescent Protein (GFP) into divergent fluorescent proteinic variants revolutionized studies on cell signaling by enabling genetically encoded biosensors [157]. These biosensors work via identifying biochemical inputs and converting them into optical outputs via fluorescence intensity, wavelength shifts, or other energy transfer mechanisms [157]. Fluorescence Resonance Energy Transfer (FRET) sensors and Single Fluorescent Protein (SFP) sensors are widely used for various applications, including assessing cellular signaling pathways, monitoring molecular interactions and protein conformations, estimating intracellular ion concentrations and studying real-time enzyme activity [158]. FRET is a distance-dependent energy transfer among a donor and an acceptor fluorophore, where excitation of the donor results into non-radiative energy transfer to the fluorophore, thus making the identification of molecular interactions and other processes possible [159]. SFP sensors are used to detect alterations in their environment, in which changes in fluorescence intensity, emission wavelength and life time reveal certain biochemical interactions/events into the studied system [158].

A common limitation of fluorescent biosensors is the pH and certain ion monitoring, whereas in peptide/protein fluorophore, poor expression and mutations (i.e., gene silencing, duplication and frameshift) can result in false-negative test results [157]. Again, in parallel with the previous reasons of false results in optical biosensing, one could argue that light source fluctuations, unsuitable wavelength of excitation light, saturated detector, light interference from background, issues with detector alignment and sensitivity, quenching (reduced fluorescence due to environmental and other conditions or cross-reactions) and overlap fluorescence, are some reasons that could trigger false results. Also, temperature, decay, background noise, photobleaching (loss of ability to emit light after prolonged exposure to excitation light), artifacts, fluorophore quantum yield as well as interferences between detectors and generally interference from excess fluorophores or other particulates can affect the final result [160,161]. As previously discussed, uneven sample or fluorophore distribution, light scattering, variations in sample absorption, turbidity and viscosity fluctuations, poor fluorophore concentrations and non-specific interactions, fluorophore aggregations, as well as erroneous excitation geometry, may be reasons for misleading results in fluorescent biosensors. Except for these issues, FRET in particular can trigger false results because of conformational changes, wrong distance among donor and acceptor, absence of overlapping spectra and low quantum yields from the donor or the acceptor [157,159].

##### 3.2.2.3. Optical Fiber Biosensors

Since the groundbreaking advancements in fiber optic technology in the 1970s, optical fibers have progressed from being simple optical transmission waveguides to sophisticated devices used in medical imaging and other fields. The key types of optical fiber sensors include interferometric sensors, fiber grating sensors, and scattering-based sensors (utilizing Rayleigh, Brillouin, or Raman scattering) [162]. Basic optical biosensors count absorbance to identify alterations in analyte concentration, while light is transmitted via an optical fiber to the sample, and the absorbed light is identified via the same or a second fiber, and the biological material of the fiber’s end interacts with the analyte [163]. Except for absorbance measurements, optical fiber biosensors can perform reflectance, fluorescence, chemiluminescence and bioluminescence measurements, as well [163]. Optical fiber sensors seem highly valuable in medical areas like medical analysis, clinical diagnostics, and biochemical detection, due to their small size, high stability, resistance to electromagnetic interference, and capability to function remotely. These biosensors can identify proteins, DNA sequences, glucose, and hormones with notable sensitivity and specificity, while protein-based detection technology is so popular in healthcare biosensors [164]. The remarkable performance of on-chip nano-photonic biosensors has been integrated with these innovative optical fibers, creating the emerging field of “Lab-on-Fiber” [164]. Optical fiber biosensors can be enzyme-, immunoassay-, nucleic acid-, and whole-cell-based and biomimetic biosensors [165].

Optical fiber biosensors are not always infallible. Similar to the other optical biosensors, they can be affected by fluctuations in temperature, humidity, pH, viscosity, as well as variations in light intensity. Additional challenges include fiber fragility, incorrect selection of wavelength and issues concerning fiber quality, signal drift, saturation of the detector, background absorbance, Raman scattering, interference because of the ambient light, poor fiber coating and optic connections, fiber contamination and sample matrix light scattering, all of which can contribute to misdiagnoses [162,166]. Also, “out-of-plane” fiber misalignment in optical fiber biosensors can lead to false results because it causes the light to propagate/scatter in unintended directions which disrupts the ability of the sensor to detect/transmit signals [167]. Two other limitations are the restricted index of refraction differences due to fiber geometry and the inability to use low/high-index materials without disrupting total internal reflection [162]. Other factors that can trigger erroneous results include fiber surface contamination or/and damage, fiber bending/twisting or/and roughness or/and wrong optic geometry, wrong fiber length, uneven sensor coating, inconsistences in the refractive index and fiber incompatible materials [164,165,168,169,170].

##### 3.2.2.4. Surface-Enhanced Raman Scattering Biosensors

Raman scattering—a phenomenon observed in 1928—comes from inelastic light interactions, which produce scattered light at various frequencies linked to molecular vibrations, yet spontaneous Raman scattering is weak, making it difficult to distinguish from intense Rayleigh scattering [171]. To enhance Raman signals, methods like resonance Raman spectroscopy and non-linear Raman methods have been developed. Furthermore, Surface-Enhanced Raman Spectroscopy (SERS) via metallic nanostructures has improved signal strength. By combining non-linear, resonant effects with metal substrates/nanoparticles boosts speed, resolution, and sensitivity, making it highly effective for bioanalysis [172]. SERS-based detection includes

(i) Direct (label-free) detection, which analyzes the Raman spectra of biomolecules on nanostructured substrates, and

(ii) Indirect detection, which uses Raman reporter-labeled SERS nanotags for enhanced sensitivity and specificity.

SERS can integrate into various biosensor types, including microfluidic technology for the detection of various analytes such as drugs, hormones, disease markers, nucleic acids, whole cells and antibiotics, as well as electrochemical, fluorescent, PCR and LFIA technology, and portable spectrometers [172].

Several factors have a negative impact on SERS biosensor results, such as the absence of uniform metallic nanostructures which can trigger inconsistent signal enhancement. Aggregated nanoparticles can affect plasmonic state and lead to reduced signal intensity. In addition, unstable laser or/and nanostructures, interference of fluorescent background sample/substrate signals, wrong laser wavelength and Rayleigh scattering overlap can all contribute to false or unreliable results in SERS biosensing applications [173,174,175,176]. The sample-relevant issues are similar with those of the previous optical biosensors, and also, variation in background signal variability, peak shifts due to certain chemical interactions, misinterpretations of overlapping and signal degradation can also give rise to false results.

##### 3.2.2.5. Photonic Crystal Biosensors

Photonic crystals are materials which own a periodically varying dielectric function in one, two, or three dimensions; the way that photons propagate in such structures is relevant to the way that electrons move in crystalline solids [177]. Photonic crystals are able to confine light efficiently and they pose tunability in their structural parameters, thus photonic crystal biosensors have been developed. Therefore, integrating holes into such structures boosts sensitivity to small refractive index alterations in a compact sensing area, allowing flexible incorporation into single-chip systems [178]. Various photonic crystal biosensors have been discussed, such as waveguides, nanoresonators, LX resonators, holes, multi-channel resonators, nanoring resonators, and fibers [178]. The photonic crystal sensors can utilize divergent methods like refractive index sensing, displacement sensing, infrared absorption and SPR, and they can broadly be classified in [179]:

(i) Waveguide-based biosensors, which can be formed via introducing a line defect in the photonic crystal, and these are used to detect bioanalytes since they control the light–matter interaction thus utilized in sensing applications;

(ii) Cavity-coupled waveguide-based biosensors, which are made by including both the point defect (cavity), which can be a micro- or nanocavity, as well as the line defect (waveguide), and they provide an excellent light–matter interaction with the analytes, thus a longer photon life time in the cavity;

(iii) Resonator-based biosensors, which are made by including cavity and waveguide along with a resonator central ring, whose shape depends on its application, and these biosensors require small analyte amounts for the purpose of sensing with high efficacy in transmission.

A reason for erroneous results in photonic crystal biosensors is the photonic fiber state (i.e., level of crystalline) [180]. Furthermore, changes in temperature, humidity, polarization, power, beam divergence and wavelength, as well as vibrations, excitation of unintended modes, stray ambient background light, silicon defect rods (due to fabrication mistakes, thermal variations or uncontrolled scattering effects), along with misalignment or spectral drift in light source, broadened peaks, surface roughness, non-uniform substrates, poor or excessive etching and the stacking of planar square-shaped inverse opal microparticles during the bioreaction and decoding process can lead to optical signal interference and misidentification, thus affecting the final result [181,182,183,184,185,186].

#### 3.2.3. Thermal Biosensors

Thermometry is about measuring temperature, with thermometers being the most common example for body or ambient temperature measurement. Nevertheless, traditional mercury thermometers have sensitivity limitations and toxicity risks. Instead, thermometric devices use sensitive thermistors to monitor the absorption or evolution of heat, which is currently applied in enzyme thermistors [187]. A “thermistor” is a type of temperature-sensitive resistor commonly used in thermal biosensors, and it alters its resistance due to changes in temperature. If present in a thermal biosensor, it can identify the heat that is produced by a bioreaction or the temperature alterations due to the interactions between analytes and receptors [188]. Thermistors can be either positive or negative temperature coefficient. So, it is evident that, even if a thermistor is often used in thermal biosensors because of its sensitivity to temperature, thermal biosensors can utilize other temperature-measuring components as well, such as thermocouples or infrared sensors [189].

Erroneous readout signals are evident in thermal biosensors and some reasons include poor thermistor calibration, thermal drift, sensor degradation, unsuitable thermistor resistance and products resistant to heat, wring thermistor placement, poor contact between thermistor and sample, lagging in response time (rapid temperature fluctuations due to environmental or other reasons), transducer temperature range exceedances or/and poor sensing range, as well as interference because of near temperature sensors [187,188,190,191,192,193,194]. Again, one could argue that extra circuitry and power supply matters, environmental factors (fluctuations in temperature, humidity, etc.) and fluctuations in thermal conductivity, can result in misdiagnoses. Insufficient insulation, sensor stress (mechanical, i.e., handling) or/and poor sensor materials, substrate properties, ununiformed temperature distribution in the specimen, temperature noise as well as poor protection, configuration or/and placement of the sensor can also trigger false results in thermal biosensors [187,188,189,190,195].

#### 3.2.4. Acoustic Biosensors

Acoustic wave sensors work via sensing alterations in the physical characteristics of an acoustic wave—such as velocity, frequency, amplitude, or phase—during interaction with a target analyte (measurand). These variations result from changes in mass, viscosity, elasticity, or conductivity at the sensing surface, enabling the detection of interactions between molecules [196,197]. Such type of biosensors can detect water pathogens, monitor environmental as well as food quality, medical diagnostics and diagnose diseases via biomolecular interactions. Therefore, in 1880, the Curie brothers discovered the piezoelectric effect, which was later named by Henkel in 1881 and first utilized by Cady in 1921.Electroacoustic biosensors detect alterations in mass density, influenced by elastic, electrical, or dielectric properties of a chemically interactive membrane which is in contact with a piezoelectric material [198]. Piezoelectric materials work as oscillators based on the piezoelectric effect, in which mechanical stress generates an electrical charge that allows easy detection of surface interactions, thus making them ideal for biosensors that detect interactions based on affinity [199]. Therefore, piezoelectric biosensors take advantage of the piezoelectric effect to convert biochemical interactions into electrical signals that operate in various modes. Through direct, label-free identification, they achieve greater sensitivity by measuring mass or mechanical alterations following analyte binding, thus enabling compatibility with a wide range of biomolecules [199]. Thus, piezoelectric biosensors are a subtype of acoustic biosensors. Common acoustic sensors can be categorized in [197]:

(i) Bulk acoustic wave sensors, which use piezoelectric materials in order to produce acoustic waves which propagate through their volume, identifying biomolecular interactions via counting alterations in wave properties like frequency, velocity, or amplitude, which happen because of variations in mass, elasticity, or viscosity at the surface of the sensor, as seen in quartz crystal microbalance and other devices,

(ii) Surface acoustic wave sensors, which on the contrary utilize piezoelectric materials to produce acoustic waves that pass along the sensor surface, identifying biomolecular interactions by counting alteration in wave properties, which occur due to alterations in mass, viscosity or elasticity at the sensor interface, as seen in molecular diagnostics, and other applications, and

(iii) Micro-/nano-electromechanical systems, which incorporate miniature electrical and mechanical components to identify biomolecular interactions through counting alterations in properties like resonance frequency, mass or displacement, which occur due to variations in elasticity, viscosity or force applied at the sensor surface, thus enabling various applications.

Acoustic biosensors are not infallible; errors in results can also occur in this case. Aside from common errors such as poor, damaged, degraded piezoelectric material, fabrication defects, saturation of the sensor and mechanical stress, other possible issues include frequency drifts, fluctuations in resonance, non-linearity in frequency responses, interfering acoustic waves and signals, signal noise, variations in surface roughness, contaminants in complex samples and unintended piezoelectric material deposition [200,201,202,203,204,205]. Some other sources of error may involve incorrect piezoelectric material and variation in its thickness, fluctuations in temperature, humidity, electric fields, pressure alterations, viscosity, density, evaporation and voltage, as well as bubble formation and air droplets pressure variations due to air currents, external vibrations, insufficient grounding and artifacts. Impedance mismatch, insufficient transducer elements alignment and also hysteresis effects can also give rise to false results [200,201,202,203,204,205]. Challenges in surface functionalization include achieving stable, reproducible attachment of biomolecules to the surface of the sensor and ensuring efficacious interactions with target analytes without compromising the performance of the sensor.

#### 3.2.5. Mechanical Biosensors

Mechanical interactions have a pivotal role in biochemistry, as they influence cellular, subcellular and molecular motility, adhesion, and transport, thus mechanical biosensing provides significant opportunities to estimate forces, displacements, and mass alterations in cellular, subcellular and molecular processes [206]. Mechanical biosensors detect the presence or/and concentration of the analytes via measuring alterations in mechanical properties like mass, force or displacement. They play a significant role in biosensing applications by converting biochemical interactions into measurable mechanical outputs. In other words, they detect these variations when target molecules interact with the surface of the sensor, and these alterations are turned into signals which offer information about the analyte’ s presence or/and concentration [206,207]. Cantilever biosensors are a type of mechanical biosensors which utilize microscopic cantilever beams to detect biological interactions via counting alterations in the beam deflection or resonance frequency. When a target binds to the surface, the interaction causes mass changes or surface stress, ultimately altering the mechanical strain or resonance frequency [208].

Both microcantilevers and micromembranes are two sensitive elements to be used in the surface stress-based biosensors; these are a type of mechanical biosensors that identify biological interactions through counting alterations in surface stress which occur after a binding between a target and the sensor surface. These biosensors track the sensor’s surface strain/deformation due to bindings that alter the surface stress, with that change to be detected and translated into a countable signal that indicates the analyte presence/concentration [209]. Gravimetric biosensors identify biochemical interactions via estimating changes in mass on a sensor’s surface, mainly by a piezoelectric/resonant system in order to monitor shifts in resonance frequency following analyte binding. These shifts are measured, and the degree of frequency change corresponds to the mass added, thereby allowing the detection of the presence/concentration of the analyte [210].

Mechanical biosensors can result in false results too; some reasons are the biological and the environmental noise (interfering vibrations) and transducer microcracks, degradation, aging, stiffness and toughness, irregular thickness, defects, thermal expansion/contraction, absence of homogeny, inhomogeneous stress distribution/residual stress and contaminations [206,211,212]. Furthermore, cantilevers’ excessive bending, overloading, instability in stress distribution, sensor surface warping, hysteresis effects, non-linear frequency responses and mechanical vibrations damping [206,211,212,213,214]. Also, one could argue that since we are referring to mechanical-based detection, fluctuations in temperature, humidity, airflow, pressure and electromagnetic interference—as well as sample viscosity, bubble formation, extremely high/low density and evaporation—can negatively impact the transducer material, thereby affecting the final result. Some other factors include the insufficient connection among the transducer and electronics, the mechanical vibrations that cause signal noise, the non-linear responses, the poor transducer integration and the possible interference amongst various sensors due to mechanical coupling issues [206,215].

#### 3.2.6. Magnetic Biosensors

In 1857, Thomson initially documented the anisotropic magnetoresistance effect, this phenomenon, and in 1996, the first magnetic bead-based bioassay was performed by Kriz et al. [216,217]. Magnetic biosensors function by detecting biological targets through measuring changes in a magnetic field—typically using magnetic nanoparticles that specifically bind to biomolecules. Once the sensor interacts with the targeted molecule, the magnetic signal is altered and then it is converted into an electrical response to be analyzed [217,218]. Such technology was first applied to detect soluble proteins, but nowadays, enzymes, DNA, pathogens and small molecules, can also be detected via magnetic biosensors [217,218]. It is clear that magnetic nanoparticles are not the actual transducers; rather, they serve as labels or probes that help generate a detectable magnetic signal after binding to the target. In reality, the real transducer is either a magnetoresistive sensor or an inductive coil, which detects changes in magnetism cause by the nanoparticles [217]. Magnetic biosensors include several types, with the most common to be the [218,219]:

(i) Anisotropic magnetoresistive biosensors, which utilize an alteration in resistance because of a magnetic field to detect biomolecules binded to magnetic nanoparticles;

(ii) Giant magnetoresistive biosensors, which resemble to the previous ones but have a larger resistance change, they have a better sensitivity and they are commonly used for applications that need high-resolution—like the detection of biomolecules in low concentrations;

(iii) Tunnel magnetoresistive biosensors, which are more sensitive than both the previous ones, because of the use of quantum tunneling effects which make them ideal for ultra-sensitive applications in biosensing;

(iv) Magnetic particle-based biosensors, which monitor alterations in the properties of the magnetic nanoparticles or magnetic beads after their interactions of certain biomolecules, usually via a sensor that is inductive or has the Hall effect.

There are several reasons for which magnetic biosensors can result in erroneous results. Magnetic nanoparticle wrong binding or/and aggregations, nanoparticle magnetic anisotropy, sensor drift, interfering materials or/and environmental noise (external magnetic fields/electronic noise) or/and low signal-to-noise ratio, sensor degradation, sensor overloading or/and uneven sensitivity, poor signal amplification, as well as fluctuations in temperature, magnetic field, magnetic nanoparticles size, and power supply can all trigger false results [217,220,221,222,223,224,225,226]. Moreover, erroneous results can occur due to magnetic hysteresis, external vibrations, sensor misalignment, sensor non-linearity, overadjustment of the baseline signal, inadequate magnetic label concentration, sensor saturation, overexposure to magnetic fields, wrong amplification circuit, interference due to capacitive binding between the sensor and other system parts, poor interaction among transducer and nanoparticles and poor transducer geometry [129,220,223,224,226].

Table 2 summarizes all potential reasons for false results in biosensors due to their transducer.

### 3.3. Current Common Biosensors Based on Their Technology

#### 3.3.1. SPR Biosensors

As was first stated in previous subsections, and since the late 1990s, SPR biosensors have been a key technology for research in the non-covalent biomolecular interactions—mostly the protein–protein and protein–small molecule ones—and practical applications in healthcare and other fields, with the method to be highly sensitive and label free. SPR biosensors are increasingly used in medical diagnostics for real-time, label-free identification of biomolecular interactions, thus enabling rapid diagnosis of health conditions such as cancer, autoimmunity and infections [227]. Optical affinity biosensors using SPR are of the most advanced label-free optical sensing technologies currently available [227]. SPR biosensors work by tracking changes in the refractive index next to a thin metal film when analyte molecules bind to immobilized biorecognition molecules. Τhis interaction is measured via an optical reader, while a sample preparation and delivery system makes it sure that the analyte can reach the surface of the sensor [227]. Some common SPR biosensors include the [228]:

(i) Conventional SPR biosensors, which use a prism-based Otto or Kretschmann configuration for the detection of refractive index alterations after the binding of the analyte,

(ii) Localized SPR biosensors, which utilize nanostructured metal nanoparticles and not a continuous metal film, which is allowed for miniaturized biosensors with high sensitivity,

(iii) Long-range SPR biosensors, which feature a thicker metal film and extra dielectric layers to boost sensitivity and extend the ranges of detection,

(iv) Waveguide SPR biosensors, which incorporate waveguides with SPR in order to enhance sensitivity and allow for more compact designs in sensors, and

(v) Graphene-enhanced SPR biosensors, which integrate layers of graphene on the metal film so as to have better selectivity and sensitivity as well.

Depending on their application, target analyte and required sensitivity, an SPR biosensor can have various bioreceptors, including antibodies, proteins, peptides, lectins, enzymes, nucleic acids and aptamers, cell membranes and whole cells [229]. Recent innovations in plasmonic nanostructures, such as nanorods, nanoshells, and hierarchical designs, enhance SPR biosensing through enhancing light–matter interactions, improving sensitivity, and enabling more accurate detection of low-abundance biomarkers. SPR biosensors face some challenges, and particularly the localized SPR sensing of membrane protein interactions is challenged by engineering metal nanoparticle systems which replicate native cellular environments with high precision, sensitivity and scalability as well. Another major issue in plasmonic sensing is the creation of amplification strategies which achieve ultra-low LoD scores while ensuring usability and stability in point-of-care devices [230]. Furthermore, another limitation is the challenge of controlling the shape and size of nanoparticles at a scalable level with precision, while ensuring reproducibility, which can trammel their widespread application [230].

Various reasons can affect the final results in SPR biosensors; for instance, it has been discussed that non-specific proteins/peptides can interact with their surface, thus giving rise to false signals [231]. Also, baseline measurements can be affected by fluctuations in the performance of the instrument. Ambient temperature and air pressure fluctuations, variations on flow rate and interfering air bubbles, contaminations in the optical components, insufficient calibration, signal-to-noise ratio and electrical/mechanical noise, poor wavelength choice, degradation of the chip surface properties, buffer impurities and interferences, cross-reactivity among channels and non-specific cross-reactions among the sample and the sensor surface can trigger false results [227,228,232,233,234,235]. Sample evaporation, viscosity, substrate variations in the sensor chip, surface fouling, external light or background substances’ interference, electrostatic or/and magnetic interference, overloading, signal saturation and of course a non-optimized time of measurement can also give rise to erroneous results [227,228,234,236,237,238].

#### 3.3.2. Microfluidic Biosensors

Microfluidic biosensors incorporate microfluidic technology with biosensing to enable direct, efficient and miniaturized bioanalysis. Such devices leverage the manipulation of small fluid volumes (mainly nanoliters but also microliters) with microchannels to enable rapid mixing, interactions and identification of the target substances. Microfluidic biosensors are principally based on laminar flow, which ensures controlled reactions as well as passive/active transport of the fluid via capillary forces and flow due to pressure, electric or kinetic mechanisms [239]. Some common applications of that technology are in electrochemical, optical and piezoelectric sensing strategies.

##### 3.3.2.1. Paper-Based Biosensors

Currently, most microfluidic diagnostic platforms are based on complex on-chip designs, fluorescence or quantum dot labeling, continuous flow and nucleic acid amplification. “On-chip” or “lab-on-a-chip” biosensors are advanced devices which incorporate biological sensing elements onto a platform of a microchip, and such sensors are utilized to identify chemical or biological analytes present in a specimen, like proteins, nucleic acids and whole cells. The main idea of on-chip biosensors is to embed the sensing ability directly onto a chip, and this makes the device compact, low-cost and efficacious [240]. Paper-based microfluidics provide a low-cost, portable solution for point-of-care diagnostics, via capillary action to move fluids, while chip-based microfluidics offers more accurate control and incorporation of complex processes; yet, it often requires expensive materials and equipment. These biosensors are of utmost diagnostic importance, with the main technology that applies into them to be the microfluidics and molecular biotechnology. Such devices are designed with numerous microchannels embedded with antibodies and antigens, as well as oligonucleotides, which enable numerous bioreactions, whereas polydimethylsiloxane seems to be the most widely used material in this technology [19]. Lab-on-a-chip biosensors can be applied in molecular biology for a faster PCR, in cell biology as it controls cells at the single-cell level randomly, and in proteomics as well [19].

LFIAS, which were somewhat discussed in previous subsections, are based on a paper-based microfluidic biosensing technology, and were first used in pregnancy tests, but the technology was publicly available under the COVID-19 rapid tests and self-tests. LFIA works in the basis of capillary-driven flow, in which a liquid mover across a nitrocellulose membrane through capillary action, and it identifies targeted biomolecules via specific probes like AuNPs or fluorescent markers [5,7,12,13].

False results in lab-on-a-chip and LFIA microfluidic biosensors may arise from various factors, including improper limits of detection (LoD) scores, fluid leakage, and clogging of microchannels. Additionally, sample evaporation, cross-contamination within microchannels, and closed microchannels can contribute to inaccuracies. Other potential causes include insufficient sample loading, issues with capillary flow or membrane porosity, as well as interferences due to the sample matrix [5,7]. Furthermore, non-uniform drying of test strips, excessive sample load, and problems with the buffering system may all affect the results. Finally, fluctuations in environmental factors such as humidity, temperature, pH, and viscosity, as well as inconsistencies in surface coatings, can further compromise the biosensor’s performance [5,7,12,13,241,242].

##### 3.3.2.2. Polymer-Based Biosensors

Microfluidic diagnostic devices that utilize magnetic methods offer a promising alternative, as they employ internal or externally applied microconductors to generate an applied current. This, in turn, facilitates the manipulation of magnetic nanoparticles or markers within microchannels using high-gradient magnetic fields. These devices also provide the advantage of labeling various biological entities via surface modification and direct binding for targeted detection [239]. Several polymers are commonly used in this technology, with the most widely employed being polydimethylsiloxane which is transparent, biocompatible and flexible for microfluidic channels, polymethyl methacrylate that is resistant and durable but more rigid, as well as the cyclic olefin copolymer that is highly transparent, does not easily absorb water and it is compatible with microfabrication and injection molding techniques [239].

Several factors can lead to false results in polymer-based biosensors due to inherent limitations in the technology. These include material inhomogeneity or roughness, which can cause inconsistent or inadequate sample flow and reactions. Other contributing factors are poor curing processes, insufficient bonding between polymer layers, delamination due to improper bonding and polymer degradation [243,244,245]. Optical interference, clogging of microchannels, inadequate polymer functionalization and poor surface treatment can mead to analyte adsorption. Additionally, insufficient polymer crosslinking, poor sealing, inadequate coating, bubble formation and trapping within microchannels, insufficient compatibility or/and hydrophobicity, poor long-term utility, as well as temperature and electroosmotic fluctuations may affect sensor performance [245,246,247,248].

#### 3.3.3. Nanotechnology-Based Biosensors

Nanotechnology-based biosensors leverage nanoscale materials like nanoparticles, nanotubes and nanowires in order to boost sensitivity, enable immediate detection and improve signal transduction, thus enabling highly accurate and miniaturized diagnostic platforms in biomedical and other applications [249].

##### 3.3.3.1. Metal Nanoparticles

Noble metal nanoparticles own distinctive physicochemical properties, such as a high surface-to-volume ratio and the ability to be easily functionalized via simple chemical methods. They possess exceptional optical and spectral characteristics that have driven the design of multifarious biosensing platforms, whereas they can act as transducers and boost or complement common utilized techniques such as localized SPR, fluorescence spectroscopy, infrared imaging, Raman spectroscopy and electrochemical approaches, LFIAs and ELISAs, thus expanding their applications in biosensing [250,251]. Metal nanoparticles can work in parallel with various bioreceptors including antibodies (AuNPs in LFIAs), aptamers, enzymes, nucleic acids as well as whole-cells and pathogens. Noble metal nanoparticles work by immobilizing bioreceptors, mediating electron transfer, catalyzing biochemical reactions, amplifying mass alteration and boosting refractive index changes [249].

Common errors leading to erroneous results in metal nanoparticle-based biosensors include aggregation caused by magnetic fields, viscosity or other factors [5]. Additionally, non-specific plasmonic effects (such as variations in pH and temperature), non-specific adsorption, (photothermal) instability and inadequate surface functionalization can negatively impact results [249,250]. Other issues include optical and other interferences, cross-reactions, signal saturation, metal ions leaching, signal quenching due to surrounding molecules, instable conjugates, ununiformed dispersion, noise due to viscous sample matrix as well as incompatibility with the overall technology of the sensor or the sample material [5,12,251,252,253].

##### 3.3.3.2. Nanostructured Surface-Based Biosensors

Nanostructured surface-based biosensors, incorporating nanowires or nanotubes, capitalize on the unique properties of 1D nanostructures, such as their high surface-to-volume rations and notable charge transport abilities, to boost the performance of biosensors. Nanowires like silicon and nanotubes like carbon nanotubes, offer a platform for immobilizing bioreceptors such as antibodies, enzymes, and nucleic acids. Such nanostructures enable highly sensitive identification because since they are able to amplify signals by mechanisms like electron transfer or even field effects [254]. They commonly function by electrochemical, piezoelectric and optical transducers, whereas they allow for real-time and accurate analyte identification in medical diagnostics and various other applications [254].

Several factors can lead to false results in biosensors utilizing nanostructure-based technology. These factors include nanostructure surface fouling, electron leakage, uneven functionalization, poor reusability, insufficient attachment of the bioreceptors in the surface of nanostructures, contamination and surface defects, cross-reactions, insufficient/non-specific bioreceptors binding, electromagnetic interference in nanowires, poor conductivity or/and signal and signal amplification, release of ions/residues from the nanostructure, noise or/and signal drift, saturation of the signal, surface oxidation, poor passivation or/and biocompatibility due to sample matrix, aggregations of nanowires/nanotubes, as well as fluctuations in nanostructure sizes (geometry, diameter or length), light, humidity, pH, ionic strength and temperature [1,255,256,257,258,259].

##### 3.3.3.3. Quantum Dots-Based Biosensors

Quantum dots are novel nanostructured luminescent materials with significant potential in various fields, particularly biology. These biosensors are capable of the immediate and accurate identification of both biological and inorganic compounds, both in living organisms and in laboratory settings. In order to improve their fluorescence characteristics and eliminate biological toxicity, quantum dots need surface modification [260]. Up to now, the initial surface modification methods include multidentate ligand attachment, sulfhydryl group coupling, amphiphilic integration of molecules, cavity-chain structures and dendrimer technology [260]. Quantum dots work via fluorescence-based detection, FRET and electrochemical or photoluminescence-based sensing, enabling medical diagnostics and other applications. Bioreceptors are the critical components which detect target molecules and they boost their performance via their unique electronic and optical properties. Some common ways through which quantum dots contribute to bioreceptors include fluorescence labeling, signal amplification, surface functionalization, as well as optical and electrochemical sensing [261,262].

Misdiagnoses can occur in quantum dot-based biosensors for reasons including fluorescence quenching (self-quenching), mix with fluorescence dye (false positive) and interferences by endogenous biological fluorescence, photobleaching, quantum dots surface defects, surface interactions with charges, quantum dots aggregations or aggregations with matrix biomolecules such as proteins, quantum dot uneven distribution, insufficient or/and non-specific binding, phototoxicity or toxicity of quantum dots that can interfere with biological systems, low target analyte levels, cross-reactions, bioreceptors clustering, as well as fluctuations in particle size, temperature, ionic strength or/and pH [260,263,264,265,266].

#### 3.3.4. Implantable Biosensors

Compared to healthy tissues, the inflamed ones, like those in cancer or diabetes, have distinct physiological alterations. Continuous monitoring of target analytes, without patient or clinician intervention, would be highly precious for monitoring trends over time. Implantable biosensors are devices partially or fully inserted into the human body for long-term monitoring of physiological conditions, offering a minimally invasive approach to track chronic diseases and assist in treatments [267]. Such biosensors are mostly electrochemical; Therefore, biotelemetry that remotes monitoring and transmission of physiological data, combined with these biosensors, enables real-time, continuous monitoring of health parameters for disease diagnosis, control and further treatment. Biosensors can also aid in drug delivery, yet there are still several biocompatibility issues [267]. Epidermal electronic systems are ultrathin, flexible devices which can integrate within the skin, thus enabling non-invasive, continuous checking of physiological signals like heart rate and skin temperature, with no discomfort or movement restrictions. Sweat sensors, like those using microfluidic systems, enable non-invasive, real-time monitoring of biomarkers like lactate, glucose, and pH, providing promising solutions for wearable health monitoring [268]. Temporary tattoo-like sensors seem a promising platform for body-compliant wearable devices, enabling non-invasive monitoring with advanced electronic incorporation and wireless transmission of data. The powering of implantable biosensors remains a major challenge. These approaches aim to provide long-lasting, autonomous power without the need for invasive maintenance. Moreover, advancements in energy harvesting techniques, including triboelectric nanogenerators and piezoelectric devices, are further enhancing the efficacy of these systems across various applications [268]. False results can be evident in implantable biosensors due to various reasons including erroneous placement or/and deformation after implantation, insufficient anchoring and dislocation of the biosensor through the body, implant misalignment, depth of implantation, surrounding tissue interferences (electroactive molecules), bacterial attachment and formation of biofilm, signal attenuation via tissues, artifacts due to body movements, biotoxicity, issues with signal-to-noise ratio due to electronic noise, poor biocompatibility, degradation, fibrous capsule formation around the biosensor or/and scar tissue formation or/and fluid accumulation, physical barriers (nearby organs, bones, etc.), as well as fluctuations in temperature, pH, viscosity and pressure [268,269,270,271,272,273,274,275,276,277].

Table 3 summarizes all potential reasons for false results in biosensors due to their technology type.

## 4. Current Common Types of AI in Biosensors in Medicine and False Results

It was discussed in previous subsections that the common AI type applied in diagnostic biosensors is machine learning, which can be further divided into supervised learning (classification and regression algorithms) and unsupervised learning (clustering algorithms and dimensionality reduction) [4,31,32].

Furthermore, some other AI types that are used for the concept of medical AI biosensors are the deep learning neutral networks and the reinforcement learning. In this section, we will briefly illustrate common AI types which are commonly integrated into biosensors or used in parallel with them and the possible reasons for false results due to such machine learning.

### 4.1. Current Supervised Learning in AI Biosensors

#### 4.1.1. Classification Algorithms

Classification has been used in both electrochemical and optical biosensors [278]. Random forest seems to be a good machine learning algorithm in AI biosensors, specifically as biological information complexity grew. Random forest was integrated to enhance data interpretation. Random forest is effective in handling with high-dimensional and noisy information, therefore important in applications like glycose monitoring as well as cancer and pathogen detection. Over time, it has been applied over time through wearable health sensors, diagnostic devices and continuous monitoring systems as well [279,280]. Random forest is an ensemble learning technique which builds several decision trees in order to predict the final result, and each tree is trained randomly on a subset of information via bootstrap sampling, in which data points are chosen by replacement. The final prediction is made by aggregating the outputs of all trees, typically through voting for classification tasks or averaging for regression tasks. This approach improves accuracy and reduces overfitting, especially compared to a single decision tree. The diversity and randomization of the trees contribute to a more robust and generalized model [281].

Furthermore, the k-nearest neighbor algorithm is commonly used for data analysis and classification—thus aiding in interpreting information for disease and biomarker detection as well as health monitoring [282,283]. This algorithm works by comparing incoming information with previously stored patterns, identifying the k-nearest data points (neighbors), and making predictions based on the average or the majority class of those data points. The algorithm does not make assumptions about the underlying data distribution. The model’s prediction is directly influenced by the distance between the test point and its nearest data points, typically measured using distance metrics [284,285].

Moreover, in AI biosensors, naïve Bayes is used for the detection of disease biomarkers, the analysis of sensor data and the distinction among health and diseased conditions according to biosensor readings. This algorithm is useful in SERS and imaging biosensors, where it helps in the classification of spectral or imaging information with high efficacy [286,287,288]. Naïve Bayes is a probabilistic ML algorithm based on Bayes’ theorem, which assumes that features are independent—under certain conditions—given the class label. Despite this assumption, the algorithm performs well in several real-world biosensing applications, particularly in processing large multivariable datasets. The algorithm also estimates the probability of a class which belongs to a given output and assigns the most possible classification; therefore, it is an essential tool for the detection of biomarkers, pathogens and diseases via biosensors [287,289,290,291].

Nevertheless, current AI algorithms are not entirely infallible. Classification algorithms may produce erroneous results due to data noise, incompleteness, imbalance in the datasets, alterations in the distribution of data (i.e., changes due to time/environmental fluctuations) inadequate selection of features and overfitting which can be evident in learning too specific patterns for the training data—thus resulting in poor performances in new data. Particularly, in random forest, some false results can be evident due to the presence of too many or too few trees thus inadequacy for real-time biosensors, overfitting because of deep trees or underfitting due to shallow trees, bootstrap sampling that may not properly represent the full dataset and can trigger biases, erroneous feature selection per tree, issues with correlated features the importance of which may be overweighed, as well as errors due to default datasets that may give misdiagnoses due to poor performance in all datasets [292,293,294,295,296,297]. On the other hand, k-nearest neighbor is highly affected by noisy data due to the absence of an internal mechanism to handle them. Ιt is also sensitive to unequal class distribution, issues with dimensionality, insufficient feature scaling that leads to some dominations in distance calculations and poor classifications. In addition, the choice of an inadequate distance metric, sensitivity due to outliers, as well as issues with near decision boundaries in which small data changes can result in fluctuations in predictions [298,299,300,301,302,303]. Finally, naïve Bayes can show incorrect final results due to various reasons, including the hypothesis of independence for all features that is impossible in biological data, the insufficient handling of linked features, the wrong estimates in prior probabilities, the zero-counts problem, the unreliability in small sample sizes or/and continuous data, imbalanced datasets and difficulties in handling real-time data [304,305,306,307,308,309].

#### 4.1.2. Regression Algorithms

Regression is considered a supervised machine learning technique which is used to model the relationship among independent variables and dependent variables in order to predict their relationship and continuous values based on them [310]. There are some common categories of regression models [310,311,312,313,314,315,316]:

(i) Univariate regression predicts one dependent variable, whereas multivariate regression predicts multiple dependent variables.

(ii) Linear regression hypothesizes a straight-line relationship among variables and can be subdivided into simple linear regression and multiple linear regression; non-linear regression captures curved relationships and incorporates methods such as polynomial and decision tree regression.

(iii) Simple regression—like simple linear regression—uses one independent linear regression to predict an outcome, whereas multiple regression—like multiple linear or multivariate multiple regression—has to do with 2+ independent variables to boost accuracy.

(iv) Bayesian regression utilizes probabilistic distributions to model uncertainty in predictions and frequentist regression that relies on fixed estimates.

(v) Regularized regression can be L1 regularization (Lasso regression), which performs feature selection through setting certain coefficients to zero and L2 regularization (Ridge regression), reducing the magnitudes of the coefficients to prevent overfitting, whereas it can also be elastic net regression that blends both L1 and L2 characteristics.

(vi) Logistic regression can be binary and multinomial (based on the number of the categories it can predict that can be 2 or 2+) and applies principles to classification problems rather than the prediction of continuous values.

(vii) Parametric regression that assumes a fixed functional forms and it can be linear, logistic and polynomial, as well as the non-parametric regression such as k-nearest neighbors, decision tree and support vector regression, which are less assumptive and can adapt to multiplexed data structures.

Some AI biosensors are based on linear regression to assess the relationship between input signals and the targeted concentration or physiological parameter of a biosensor. These biosensors have a wide range of applications, including glycose monitoring as well as in electrochemical, optical and wearable technologies for the detection of diseases, pathogens and various biomarkers [317,318]. Also, polynomial regression in AI biosensors is used to model non-linear relationships among sensor signals and biomarker concentrations, estimating complex biological interactions. Polynomial regression is applied in fluorescence-based sensors, enzyme kinetics modeling, and non-linear electrochemical responses for better detection [319,320]. Some other regression types have also been used [32].

Again, there exist some reasons for false results in regression models. In general, overfitting can occur due to noise capturing rather than the real pattern, while underfitting results from the simplicity of the model, which fails to capture critical data relationships [321,322]. Other challenges include multicollinearity which is highly possible in physiological data, potential outliers, data leakage, inadequate data and insufficient quality as well as incorrect assumptions [321,322,323]. In particular, linear regression can give false results due to violation of linearity, autocorrelation, multicollinearity, homo- or heteroscedasticity, bias in omitted variables, the presence of too many outliers, as well as certain values that are highly influential [324,325,326]. Polynomial regression can save similar reasons for erroneous results, including issues with extrapolation, sensitive data, unreal curve fluctuations due to high-degree polynomials (oscillation), higher computational cost and difficult interpretations as well [327,328]. Moreover, multivariate regression can be affected by additional issues, including high dimensionality, missing data in some variables, variance and unstable model performance. Noise from irrelevant predictors, issues with feature scaling as well as complexity since more independent variables need higher processing power and all these can also impact the model [329,330,331]. Lasso regression may be subject to erroneous results due to over-regularization, multicollinearity, very small dataset, unsuitable λ, presence of outliers, non-linearity in relationships, non-scaled features as well as random features, and data leakage, as well [332,333,334,335]. Moreover, misdiagnoses may occur in AI biosensors empowered with ridge regression due to similar reasons like over-regularization, wrong λ, non-linearity, presence of outliers, unsuitable scaling of features, leakage of data, very small dataset and inadequate initial hypotheses [336,337,338].

### 4.2. Current Unsupervised Learning in AI Biosensors

#### 4.2.1. Clustering Algorithms

Clustering is a critical tool for exploratory analysis, and its algorithms are unsupervised learning techniques that group similar data points based on their characteristics. The goal is to identify inherent patterns or structures in the data with no need for labeled outcomes. Hierarchical clustering is an unsupervised machine learning technique which groups data points in a cluster hierarchy according to their similarity via either a bottom–up or a top–down way. This method involves progressively merging or splitting data points by their similarity via distance metrics to aid the process, and visualizing the results via a dendrogram to identify data hierarchical structure [339]. This method can be used in biosensors to categorize and analyze multiplexed biological data like gene expression or sensor signaling via grouping same patterns to identify diseases or/and monitor biological processes. Common types of biosensors include the immunosensors, enzyme biosensors, optical biosensors and wearable biosensors [340,341].

K-means clustering analysis is another unsupervised machine learning algorithm that groups data points in K different clusters according to their similarities and therefore the variance within each cluster is minimized. The key principles include the selection of the number of clusters (K), assigning data points to their nearest centroid and updating the last ones iteratively until convergence [342]. This method is commonly used in biosensors for the recognition of patterns and diseases, as well as for the classification of biomarkers, thus enabling adequate analysis of complex biodata. Some biosensor types in which it is applied are the immunosensors, the enzymatic and the electrochemical biosensors [343,344].

Gaussian mixture model clustering may provide better clustering, and it is a probabilistic clustering algorithm which hypothesizes that data are produced by a mixture of several Gaussian distributions from which each one represents a cluster [345,346]. This clustering method is used in the analysis of biosignals and in medical diagnostics, where data distributions are multiplexed and they overlap [347,348].

In parallel with previous analyses methods, clustering analysis can trigger some false results due to various reasons including insufficient or wrong number of clusters, scaling issues, noise in data, incorrect distance metric, imbalanced cluster sizes and absence of ground truth. Hierarchical clustering analysis can result in misdiagnoses due to incorrect linkage method and insufficient separation of clusters, outliers, impractical clustering due to large datasets, wrong cut-off of the arbitrary cluster, scaling issues and problems arising from imbalance in cluster sizes [349,350,351]. In addition, unnecessary features affecting the formation of clusters, improper validation, as well as noise overfitting [349,350,351]. Again, k-clustering algorithm may produce false results due to incorrect cluster number, outliers, hypothesized spherical clusters, random initialization issues, issues with scalability and feature, data high-dimensionality, imbalanced cluster sizes, absence of probabilistic assignments and ignorance of relationships among features [342,352,353,354]. Finally, the Gaussian mixture model analysis is affected by wrong components’ number choice, incomplete data, sensitivity to initialization [355,356]. Other challenges include overlapping clusters, wrong covariance matrix selection, high-dimensionality or high computational cost, absence of normality, issues with the expectation-maximization algorithm as well as challenges with the interpretation of the results [355,356,357,358].

#### 4.2.2. Dimensionality Reduction Algorithms

Dimensionality reduction algorithms are methods used to reduce the number of variables in a dataset while preserving as much relevant information as possible. These techniques help improve computational efficiency, reduce noise, and enhance the visualization of high-dimensional data. These techniques can be

(i) Linear methods, which reduce dimensions via applying linear transformations to maintain variance/class separability—they include principal component analysis, linear discriminant analysis and singular value decomposition;

(ii) Non-linear methods, which capture complex data structures through maintaining local/uniform relationships,—they include t-distributed stochastic neighbor embedding, isomap, autoencoders and uniform manifold approximation and projection.

Principal component analysis is a common method that transforms high-dimensional data in low-dimensional space but it maintains the maximum variance. The basic principles of this method include data transformation in a new set of orthogonal components which maximize variance, using eigenvalues and eigenvectors to reveal the critical values and lessening dimensionality and essential data for analysis [359,360]. This is the most common dimensionality reduction method which has been used in biosensors for such purposes, enabling the identification of features and key patterns for more accurate detection of diseases and bioanalysis [361,362,363,364].

False results in AI biosensors due to principal component analysis can occur for reasons including insufficient data preprocessing, overfitting, underfitting, outliers, absence of linear relationships, feature redundancy or irrelevant features, missing entries, non-Gaussian distributions, loss of critical data, as well as small inadequate datasets [359,360,365,366,367,368].

### 4.3. Current Deep Learning Neutral Networks in AI Biosensors

Deep learning neutral networks are some computational models, which consist of multiple layers of interconnected neurons that process and learn from large data amounts. They are based on layered architectures that include input, hidden and output layers, back learning propagation, activation functions for non-linearity and extra algorithms for optimization such as gradient to adjust weights [369,370]. Such networks are commonly used in image recognition, natural language processing, healthcare and diagnostics, autonomous systems as well as in AI biosensors—where they aid in recognition of complex patterns and decision making [370]. AI biosensors can use deep learning neutral networks in order to analyze and further interpret complexed biological data, therefore enabling accurate detection, classification and disease prediction. These networks can learn intricate patterns from the sensor’s information, like levels of biomarkers, gene expression and electrocardiogram signals, thus offering better diagnostic accuracy and efficacy.

Although deep learning neutral networks can provide precise results in AI biosensors, erroneous results can occur in certain cases for some reasons. Poor training or/and noisy data, overfitting or underfitting, training data mislabeling, data imbalance, wrong hyperparameter tuning, data leakage, feature oversimplification, poor model testing, inconsistent collection of data, poor validation and improper regularization methods can all affect the final result and trigger misdiagnoses [371,372,373,374,375].

### 4.4. Current Reinforced Learning in AI Biosensors

Machine learning includes also reinforced learning, in which an agent learns how to make decisions after its interaction with an environment to maximize cumulative rewards. Machine learning operates via trial and error, where the environment gives feedback to the agent in the form of rewards or penalties, thus adjusting its actions in time in order to enhance its performance [376]. In biosensors, reinforcement learning can be used to optimize decision-making processes, including improvements in real-time monitoring, disease and molecular diagnostics, as well as treatment recommendations based on sensor data. Reinforcement learning can also be applied to various biosensors, such as glucose sensors, wearable devices, electrocardiograms, and other biosensors for disease detection, thereby enhancing the accuracy of predictions and improving the adaptation to changing biological conditions [376,377].

Reinforcement learning can produce false results for several reasons such as inadequate exploration, environmental noise or irrelevant features that can give rise to noisy data, underfitting or overfitting, poor preprocess of the data or/and labeling, inappropriate architecture of the model, data imbalance or/and leakage, insufficient hyperparameter tuning, over-simplified features, poor testing, variable collection of information, inadequate cross-validation as well as loss of regularization [376,378,379,380].

Table 4 summarizes all potential reasons for false results in AI biosensors due to AI.

## 5. The Expert’s Opinion and the Future Perspectives

The aim of this critical state-of-the-art review article is to present the common reasons behind false results in current biosensors, as well as the reasons that cause the AI integrated into biosensors to yield further misdiagnoses. As such, not only can the scientific community prevent possible erroneous results, but also enhance biosensor accuracy, while future directions for improved AI implementation in this field (for preventing some false results) are concluded.

Since initial biosensors have several limitations—primarily related to their bioreceptor, transducer, and underlying technology—applying AI without addressing these constraints can significantly amplify the risk of misdiagnoses. If these limitations are overlooked, the integration of AI may lead to severe diagnostic errors rather than improving accuracy. Even if such initial biosensor limitations are solved, one must bear in mind that AI has its limitations as well, and as the famous George Box said, “essentially all models are wrong, but some are useful” [356]. Indeed, AI technologies can significantly impact various types of biosensors by improving their accuracy, efficiency, and adaptability. For enzyme-based biosensors, AI can optimize reaction rates and predict enzyme behavior in different conditions, enhancing sensor performance. In antibody-based and protein-based biosensors, AI helps identify optimal binding patterns, ensuring more reliable detection of target molecules. AI also aids DNA aptamer and whole-cell-based biosensors by analyzing complex biological data, enabling better biomarker identification and disease diagnosis. Additionally, for electrochemical, optical, thermal, acoustic, mechanical, and magnetic biosensors, AI algorithms can refine signal processing, noise reduction, and interpretation of various sensor outputs, resulting in more accurate and faster diagnostic results.

AI-based biosensor data interpretation enhances accuracy and efficiency by analyzing complex patterns and optimizing decision making, but challenges include the need for large, high-quality datasets and ensuring algorithm transparency. Furthermore, it has been discussed that not only clinical samples but also diseases—from common flu to cancer—are heterogeneous due to various reasons including gender, age, genetic variations, lifestyle factors, environmental influences, and underlying health conditions [76]. This heterogeneity challenges both traditional biosensors and AI-based diagnostic systems, since variations in biological samples can affect the accuracy and reliability of all detection methods. Moreover, it is not sufficient to focus solely on false positives and false negatives, as their rates are random and can vary depending on numerous factors, such as heterogeneity not only in clinical but also in population-based samples. Therefore, not only clinical samples are random but also the populaces are random and heterogeneous as well. Despite these diagnostic challenges, AI has the potential to overcome many of them by leveraging advanced data processing, pattern recognition, and adaptive learning techniques. At first glance, one might suggest that various biosensor AI algorithms should be designed and combined to account for all these parameters. Additionally, a biosensor should capture all relevant physiological data from an individual and process it using advanced AI algorithms to generate accurate final results. AI biosensors are only valuable when the data they generate can be effectively utilized, and understanding the meaning of the data is crucial. AI data processing involves learning from data, analyzing it properly, drawing correct conclusions, and detecting misinterpretations. One could argue that in order to ensure accurate detection and diagnosis of most biomarkers, the levels of heterophile antibodies and paraproteins—previously discussed as factors prone to causing misdiagnoses—must be carefully assessed and incorporated into AI algorithms. One could also argue that AI should not solely rely on chemical information but also incorporate other parameters, such as thermal data, as factors like temperature can significantly affect the results. Therefore, an integrated algorithm must evaluate information from all sensors within the AI biosensor to provide a comprehensive and accurate final result. So, future biosensors may be designed to integrate sensors for biological, chemical, optical, magnetic, mechanical, and other relevant data, with future enhanced AI algorithms combining all these inputs to provide a comprehensive and accurate final result. Such combination should be based on several data included in Figure 1 as well as in Table 2 and Table 3. However, it is important to note that while accuracy is crucial, more precise algorithms tend to be slower, meaning that rapid results may not hold as much value in the context of comprehensive AI biosensors [307]. So, after taking into account Figure 1 and Table 3 and Table 4 and achieving a combined biosensor result, AI output must be assessed in parallel with its limitations as seen in Table 4. Yet, Figure 1 as well as Table 2, Table 3 and Table 4 present only some of the reasons for possible false results in common biosensors based on their bioreceptor, transducer and technology, as well as the commonly used AI in biosensors, thus one must take into account the limitations of the specific biosensor and AI type used in his case, in order to draw further conclusions after the final result. Further, regulatory hurdles and standardization challenges can arise from the need for non-stop and precise validation, cross-platform compatibility, and meeting safety and accuracy standards in different healthcare environments.

Finally, while AI biosensors—which step away from traditional designs—may exhibit enhanced reliability and reduced error rates, it is essential to remember that even conventional biosensors, when empowered by AI, can produce more refined and comprehensive outcomes. The integration of AI allows classical biosensors to overcome inherent limitations discussed in this unique state-of-the-art review article, by adapting their responses based on real-time data. Such fusion not only boosts accuracy but also enables a more spherical approach to results, providing a broader understanding of the biomolecular environment. As a result, both traditional and innovative biosensor models equipped with AI hold great promise for improving diagnostics, monitoring, and precision in various applications.

## 6. Conclusions

In conclusion, while false results in biosensors and AI methods remain a significant concern, addressing their root causes can greatly enhance diagnostic accuracy. By refining AI algorithms and integrating diverse sensor data, many challenges posed by heterogeneity and variability in clinical and population samples can be mitigated. With ongoing advancements in AI and biosensor technology, the potential for more reliable, efficient, and personalized diagnostic tools is highly promising. Ultimately, the transformative potential of biosensors in revolutionizing healthcare and diagnostics cannot be overstated, offering new frontiers for precision medicine and patient care.

## Figures and Tables

**Figure 1 diagnostics-15-01037-f001:**
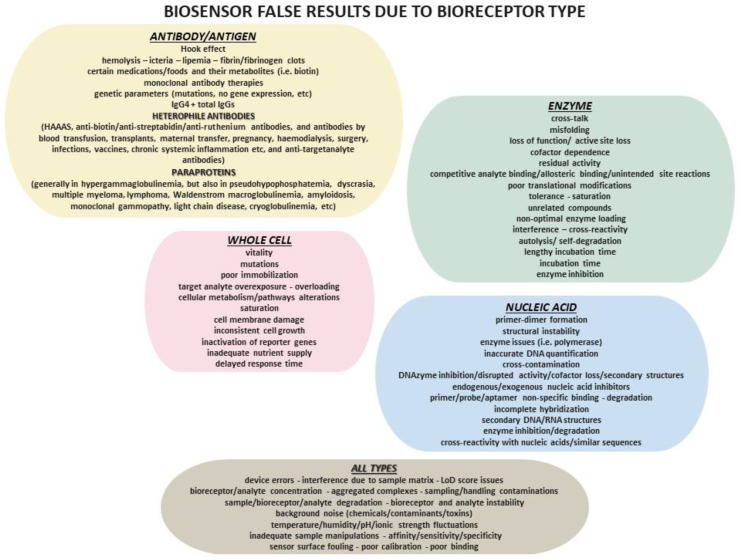
Biosensor false results due to bioreceptor type.

**Table 1 diagnostics-15-01037-t001:** Table 1 summarizes the generations of biosensors and compares them.

Generation	Key Features	Examples	Technological Advancements
1st	basic integration of biological and physicochemical components for basically qualitative/semi-quantitative measurements	enzyme-based biosensors	simple early-stage biosensors on direct biological interactions
2nd	better sensitivity and selectivity with a focus on quantitative measurements	immunosensors, DNA sensors	usage of more refined biological recognition elements like antibodies
3rd	incorporation of advanced materials and nanotechnology, with miniaturization and higher specificity	nanobiosensors, lab-on-a-chip	boosted sensor surface chemistry or microfluidics for faster results
4th	real-time detection with wireless communication and incorporation with mobile devices and cloud technologies	wearable biosensors, implantable sensors	design of real-time continuous monitoring systems which provide real-time transmission of data
5th	fully incorporated, smart biosensors with AI and ML for predictive analytics	AI-powered diagnostic sensors	AI-driven data analysis for personalized medicine and diagnostics

**Table 2 diagnostics-15-01037-t002:** Biosensor false results due to transducer type.

Biosensor Method	Biosensor Transducer Type	Reasons for Possible False Results
electrochemical	voltametric	– residual currents (due to charging effects) – self-assembled monolayers (poor conductivity, defect formation, desorption during redox, low wettability, low corrosion resistance) – graphene (low dispersibility in water, tendency to agglomerate) – carbon nanotubes (non-biodegradability, tendency to aggregate, poor solubility, challenges with reproducibility and impurities) – metal/metal oxide nanoparticles (agglomeration, oxidation, corrosion, altering protein stability, structure, and function) – quantum dots (low stability, high power consumption) – magnetic nanoparticles (low dispersion in aqueous solutions, chemical instability) – peptide analysis issues (requirement for high negative potential, sensitivity variations in catalytic hydrogen evolution, interference from donor chemical species, variability in reduced transition metal complexes, limited electroactive amino acids, irreversible and multistep nature of amino acid oxidation, reliance on peak position, height, and shape) – sample/experimental conditions (low analyte presence, short deposition time, variations in pH and ionic strength, poor mass transport, signal suppression by complex matrices, inefficient range, passivation of electrodes, competitive adsorption) – instrumental/environmental factors (non-specific redox reactions, contaminants and secondary species, electrode fouling and instability, redox cycling, instrumental drift, incomplete rinsing, cross-reactivity, electromagnetic interference)
	amperometric	– electrode and signal interference issues (interference from similar electroactive species, non-specific adsorption, residual currents) – environmental and external interference (external noise, matrix composition) – sensor performance and stability issues (electron fouling, instrumental drift, electrode fouling and instability, redox cycling, incomplete rinsing) – electrode potential issues (overly high electrode potential) – instrumental/environmental factors (contaminants, cross-reactivity, electromagnetic interference)
	impedimetric	– surface preparation and pretreatment issues (variations in electrode pretreatments, inconsistent pretreatment effects, inefficient surface preparation, poor pretreatment and cleaning, incomplete blocking of the electron surface, fouling of the electron surface) – surface functionalization issues (incorrect chemical coupling, incorrect ligand orientation on the sensor surface) – sensor performance and calibration issues (poor repeatability of inter-electrode measurements, large standard deviations in sensor responses, poor calibration of the sensor, variances in sensor fabrication, regeneration issues, leakages and formation of liquid or air bubbles) – environmental and external interference (variations in temperature and p, noise due to external electrons, power fluctuations, interference by external electronic devices) – chemical and biological interference (cross-reactions, impurities and contaminants, unintended redox reactions on the electrode surface)
	potentiometric	– environmental and external interference (fluctuations in pH, oxygen, temperature, and humidity, electrostatic and electromagnetic interference) – sample and matrix interference (cross-reactions with other non-target analytes: ammonia, lipids, sample contaminants, surfactants, detergents, gas permeability, acidity, solubility, inward and outward flux species, alterations in internal filling electrolyte composition, equilibrium selectivity, diffusion-layer effects in solid-state ISES, membrane ion-exchange competition, modified diffusion processes, membrane degradation, loss of biocatalytic stability/inactivation, sample activators or inhibitors) – electrode and sensor issues (imbalance, unstable electrodes: especially the reference electrode causing drift, improper calibration, sensors aging/degradation, improper storage conditions, manufacturing issues) – photoactive material and light source issues (in LAPS) (poor light source and intensity, erroneous light wavelength, unsuitable thickness of the insulating layer, low-sensitive photoactive material, light scattering) – MIP-based potentiometric biosensor issues (poor/wrong polymer template selection, polymerization issues and conditions, failure in removing the template, poor binding interactions, polymer instability, cross-reactions, interferences, overloading: high analyte concentrations, erroneous template-to-monomers ratio, low analyte concentrations, matrix issues: viscosity) – photoelectrochemical potentiometric biosensor issues (light source malfunction, low/high light exposure, reversely exposure to external light: high intensity/analyte concentration saturation, photoactive material degradation, inconsistent coverage, poor charge carrier mobility, noise, low photoelectrode sensitivity, unstable/poor electrical contact, delayed response time)
	conductometric	– ingredient and composition issues (wrong ratio of ingredients, absence of active substances) – environmental and sample interference (fluctuations in temperature, pH, and ionic strength, chemical reactions, ion interferences, contaminations) – sensor and electrode performance issues (sensor drift, electrode fouling, poor enzymatic activity, low sample volume, electrochemical noise, overload, poor membrane/coating)
optical	colorimetric	– environmental and sample interference (changes in surrounding light, background color interference, variations in humidity, temperature, turbidity, viscosity, ionic strength, and pH, oxidation effects altering chromophore states and inducing color variations, interferences with colored compounds) – sensor and detection issues (inconsistent light source, light-scattering nanoparticles, autofluorescence due to microfluidic chip in polymeric devices, fluorescence due to food/drug products, indicator precipitation) – reaction and analytical issues (inhomogeneous reactions, ununiformed dyeing and color distribution, chromatic aberration, aggregated nanoparticles causing artifacts) – contamination and interference (chemical contamination: endogenous or exogenous factors, interference and inhibitors, photobleaching, photochemical destruction)
	fluorescent	– environmental and sample interference (ph variations, ion monitoring issues, temperature, turbidity, viscosity fluctuations, variations in sample absorption, excess fluorophores or other particulates) – sensor and detection issues (light source fluctuations, unsuitable wavelength of excitation light, saturated detector, light interference from background, issues with detector alignment and sensitivity, background noise, quenching and overlap fluorescence, fluorophore quantum yield, interference from detectors) – fluorophore and reaction issues (poor expression and mutations: gene silencing, duplication, frameshift, uneven sample or fluorophore distribution, fluorophore aggregations, non-specific interactions, poor fluorophore concentrations, erroneous excitation geometry) – FRET issues (conformational changes, wrong distance between donor and acceptor, absence of overlapping spectra, low quantum yields from donor or acceptor) – other analytical issues (photo-bleaching, artifacts)
	optical fiber	– environmental and sample interference (fluctuations in temperature, humidity, pH, viscosity, light intensity, ambient light interference, sample matrix light scattering) – sensor and detection issues (signal drift, saturation of the detector, background absorbance, Raman scattering, poor fiber coating, poor optic connections) – fiber and structural issues (fiber contamination, fiber surface contamination or damage, fiber bending, twisting, or roughness, wrong optic geometry, incorrect fiber length, uneven sensor coating, inconsistencies in refractive index, fiber incompatible materials) – misalignment and propagation issues (out-of-plane fiber misalignment, restricted index of refraction differences, inability to use low/high-index materials without disrupting total internal reflection)
	surface-enhanced Raman scattering	– nanostructure and signal issues (absence of uniform metallic nanostructures, aggregated nanoparticles affecting plasmonic state and signal intensity, unstable laser or nanostructures) – interference and background issues (interference of fluorescent background sample/substrate signals, variation in background signal variability, peak shifts due to chemical interactions, misinterpretations of overlapping signals, signal degradation) – excitation and detection issues (wrong laser wavelength, Rayleigh scattering overlap)
	photonic crystal	– structural and material issues (photonic fiber state, Si defect rod due to fabrication mistakes, thermal variations, or uncontrolled scattering effects, surface roughness, ununiformed substrates, stacking of planar square-shaped inverse opal microparticles during bioreaction and decoding) – environmental and optical interference (changes in temperature, humidity, polarization, power, beam divergence and wavelength, vibrations, stray ambient background light) – alignment and signal detection issues (misalignment/spectral drift in light source, broadened peaks, excitation of unintended modes)
thermal		– sensor and calibration issues (poor thermistor calibration, thermal drift, sensor degradation, unsuitable thermistor resistance, wring thermistor placement, poor contact between thermistor and sample, lagging response time, exceedance of transducer temperature range, poor sensing range) – environmental and external interference (fluctuations in temperature, humidity, thermal conductivity fluctuations, near temperature sensors interference, extra circuitry and power supply issues) – mechanical and structural issues (insufficient insulation, sensor stress due to mechanical handling, poor sensor materials, substrate properties, ununiformed temperature distribution in the specimen, temperature noise, poor protection, configuration or placement of the sensor)
acoustic		– material and fabrication issues (poor/damaged/degraded piezoelectric material, fabrication defects, wrong piezoelectric material, variation in piezoelectric material thickness, unintended piezoelectric material deposition) – signal and resonance issues (saturation of the sensor, frequency drifts, fluctuations in resonance, non-linearity in frequency responses, interfering acoustic waves and signals, signal noise) – environmental and external interference (fluctuations in temperature, humidity, electric fields, pressure alterations, viscosity, density, evaporation, voltage, bubble formation, air droplets pressure variations, external vibrations, air currents) – sensor alignment and mechanical issues (insufficient grounding, insufficient transducer elements alignment, impedance mismatch, hysteresis effects) – sample and contamination issues (contaminants in complex samples, variations in surface roughness)
mechanical		– material and structural issues (biological and environmental noise, transducer microcracks, degradation, aging, stiffness and toughness, irregular thickness, defects, thermal expansion/contraction, absence of homogeneity, inhomogeneous stress distribution, residual stress, contaminations) – sensor mechanical issues (cantilever excessive bending, overloading, instability in stress distribution, sensor surface warping, hysteresis effects, non-linear frequency responses, mechanical vibrations damping) – environmental issues and external interference (fluctuations in temperature, humidity, airflow, pressure, electromagnetic interference, sample viscosity, bubble formation, extremely high/low density, evaporation)
magnetic		– nanoparticle issues (magnetic nanoparticle wrong binding, nanoparticle aggregation, magnetic anisotropy) – sensor and signal issues (sensor drift, sensor degradation, sensor overloading, uneven sensitivity, poor signal amplification, sensor saturation, overexposure to magnetic fields, wrong amplification circuit, overadjustment of the baseline signal) – environmental and external interference (external magnetic fields, electronic noise, low signal-to-noise ratio, fluctuations in temperature, magnetic field, magnetic nanoparticle size, power supply, external vibrations) – sensor alignment and interaction issues (magnetic hysteresis, sensor misalignment, sensor non-linearity, capacitive binding interference, poor interaction between transducer and nanoparticles, poor transducer geometry, inadequate magnetic label concentration)

**Table 3 diagnostics-15-01037-t003:** Biosensor false results due to technology type.

Biosensor Technology Type		Reasons for Possible False Results
SPR		– surface interaction issues (non-specific proteins/peptides interaction with the surface, non-specific cross-reactions among the sample and the sensor surface) – instrumental and calibration issues (fluctuations in instrument performance, insufficient calibration, signal-to-noise ratio, electrical/mechanical noise, poor wavelength choice, degradation of chip surface properties) – environmental and sample issues (ambient temperature fluctuations, air pressure fluctuations, variations in flow rate, interfering air bubbles, sample evaporation, viscosity, substrate variations in the sensor chip, surface fouling) – interference and contamination (contaminations in optical components, buffer impurities and interferences, cross-reactivity among channels, external light interference, background substances interference, electrostatic or magnetic interference)
microfluidic	paper based	– fluid and sample handling issues (improper LoD scores, fluid leakages, clogging of microchannels, sample evaporation, microchannels cross-contamination, closed microchannels, insufficient sample load, capillary flow issues, membrane porosity) – sample matrix and test strip issues (interferences due to sample matrix, ununiformed test strip drying, excessive sample load, issues with the buffering system) – environmental and external factors (fluctuations in humidity, temperature, pH, viscosity, surface coatings)
	polymer based	– material and flow issues (inhomogeneity, roughness of material, inconsistent or inadequate sample flow/reactions, clogging of microchannels, bubble formation and trapping within microchannels) – polymer layer issues (poor curing processes, insufficient bonding among polymer layers, delamination, insufficient polymer crosslinking, poor sealing, inadequate coating) – surface and functionalization issues (poor surface treatment, resulting in analyte adsorption, inadequate polymer functionalization) – material compatibility and stability (insufficient compatibility, poor long-term utility, hydrophobicity issues, polymer degradation) – environmental factors (temperature fluctuations, electroosmotic fluctuations)
nanotechnology	metal nanoparticles	– aggregation and dispersion issues (aggregation due to magnetic fields, viscosity or other causes, ununiformed dispersion) – surface functionalization and stability issues (inadequate surface functionalization, photothermal instability, instable conjugates) – plasmonic and optical effects (non-specific plasmonic effects due to pH, temperature, etc., optical and other interferences, cross-reactions, signal saturation, signal quenching) – contamination and chemical issues (metal ions leaching, non-specific adsorption) – environmental and sample matrix factors (noise due to viscous sample matrix, incompatibility with sensor technology or sample material)
	nanostructured surface based	– surface and functionalization issues (nanostructure surface fouling, uneven functionalization, poor reusability, insufficient attachment of bioreceptors, contamination, surface defects, cross-reactions) – bioreceptor binding issues (insufficient/non-specific bioreceptor binding) – electrical and signal issues (electron leakage, poor conductivity, signal drift, signal amplification issues, saturation of the signal, noise) – nanostructure-specific issues (aggregations of nanowires/nanotubes, release of ions/residues from the nanostructure, surface oxidation) – environmental and sample matrix factors (electromagnetic interference with nanowires, fluctuations in nanostructure sizes, geometry, diameter or length, light, humidity, pH, ionic strength, temperature)
	quantum dots based	– fluorescence issues (fluorescence quenching, mix with fluorescence dye, photobleaching, interference by endogenous biological fluorescence) – quantum dot-specific issues (quantum dot surface defects, surface interactions with charges, quantum dot aggregations, aggregations with matrix biomolecules such as proteins, quantum dot uneven distribution, bioreceptor clustering) – binding and reaction issues (insufficient or non-specific binding, cross-reactions) – environmental and sample factors (low target analyte levels, fluctuations in particle size, temperature, ionic strength, pH) – toxicity issues (phototoxicity, toxicity of quantum dots interfering with biological systems)
implantable		– placement and physical issues (erroneous placement, deformation after implantation, insufficient anchoring, dislocation, implant misalignment, depth of implantation, physical barriers such as nearby organs or bones) – biological and tissue interference (surrounding tissue interferences, electroactive molecules, bacterial attachment, biofilm formation, fibrous capsule formation, scar tissue formation, fluid accumulation) – signal issues (signal attenuation via tissues, artifacts due to body movements, issues with signal-to-noise ratio, electronic noise) – biocompatibility and toxicity (biotoxicity, poor biocompatibility, degradation) – environmental fluctuations (fluctuations in temperature, pH, viscosity, pressure)

**Table 4 diagnostics-15-01037-t004:** AI biosensor false results due to AI.

Biosensor Algorithm Used		Reasons for Possible False Results
Supervised learning	classification	– general issues for (data noise, data incompletion, imbalance in datasets, alterations in data distribution due to time/environmental fluctuations, inadequate selection of features, overfitting, poor performance on new data) – random forest issues (too many or too few trees, inadequacy for real-time biosensors, overfitting due to deep trees, underfitting due to shallow trees, bootstrap sampling not representing the full dataset, erroneous feature selection per tree, overweighting correlated features, errors due to default datasets) – k-nearest neighbor issues (noisy data, absence of internal noise handling, unequal class distribution, dimensionality issues, insufficient feature scaling, inadequate distance metric choice, sensitivity to outliers, near decision boundaries causing fluctuations in predictions) – naïve bayes issues (independence assumption of features, insufficient handling of linked features, wrong estimates in prior probabilities, zero-counts problem, unreliable small sample sizes, imbalanced datasets, difficulties in handling real-time data)
	regression	– general issues (overfitting, underfitting, multicollinearity, outliers, data leakage, inadequate data/insufficient quality, wrong assumptions) – linear regression issues (violation of linearity, autocorrelation, multicollinearity, homoscedasticity/heteroscedasticity, bias in omitted variables, presence of too many outliers, highly influential values) – polynomial regression issues (extrapolation, sensitive data, unreal curve fluctuations, higher computational cost, difficult interpretations) – multivariate regression issues (high dimensionality, missing data, variance issues, noise due to irrelevant predictors, feature scaling, complexity) – lasso regression issues (over-regularization, multicollinearity, very small dataset, unsuitable λ, presence of outliers, non-linearity, non-scaled features, random features, data leakage) – ridge regression issues (over-regularization, wrong λ, non-linearity, outliers, unsuitable scaling of features, data leakage, very small dataset, inadequate initial hypotheses)
unsupervised learning	clustering	– general issues in clustering analysis (insufficient or wrong number of clusters, scaling issues, noise in data, incorrect distance metric, imbalanced cluster sizes, absence of ground truth) – hierarchical clustering issues (incorrect linkage method, insufficient separation of clusters, outliers, impractical clustering due to large datasets, wrong cut-off of arbitrary cluster, scaling issues, imbalance in cluster sizes, unnecessary features affecting cluster formation, improper validation, noise overfitting) – k-clustering algorithm issues (incorrect cluster number, outliers, hypothesized spherical clusters, random initialization issues, issues with scalability and feature, high-dimensionality, imbalanced cluster sizes, absence of probabilistic assignments, ignorance of relationships among features) – Gaussian mixture model analysis issues (wrong components’ number choice, incomplete data, sensitivity to initialization, overlapping clusters, wrong covariance matrix selection, high-dimensionality, high computational cost, absence of normality, issues with the expectation-maximization algorithm, challenges with result interpretation)
	dimensionality reduction (PCA)	– insufficient data preprocessing, overfitting, underfitting, outliers, absence of linear relationships, feature redundancy or irrelevant features, missing entries, non-Gaussian distributions, loss of critical data, small inadequate datasets
deep learning neutral networks		– poor training or noisy data, overfitting or underfitting, training data mislabeling, data imbalance, wrong hyperparameter tuning, data leakage, feature oversimplification, poor model testing, inconsistent collection of data, poor validation, improper regularization methods
reinforced learning		– inadequate exploration, environmental noise or irrelevant features, underfitting or overfitting, poor preprocessing of the data or labeling, inappropriate architecture of the model, data imbalance or leakage, insufficient hyperparameter tuning, over-simplified features, poor testing, variable collection of information, inadequate cross-validation, loss of regularization

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
