# Peer review of "Biosensors, Artificial Intelligence Biosensors, False Results and Novel Future Perspectives"

_diagnostics, 2025, doi:10.3390/diagnostics15081037_

Round 1

Reviewer 1 Report

Comments and Suggestions for Authors

Dear Authors,

Your article, "Medical Diagnostic Biosensors and Artificial Intelligence: Bio-Sensors and False Results," is a review of existing medical diagnostic biosensors and their applications. It is of interest to readers.

The review covers the history of the development and application of medical diagnostic biosensors, as well as the current state of the industry on this subject. The article provides fundamental definitions and a thorough examination of biosensor types. A notable strength of the article is its exploration of artificial intelligence's integration into the field of diagnostics, accompanied by a discussion of the associated challenges. This article is of particular significance in this regard, as it underscores the critical issue of human-technology interaction and emphasizes the necessity for systematic studies. These studies serve as a call to the scientific community to conduct a comprehensive examination of the application of new technologies and their enhancement, particularly in domains related to healthcare, where diagnosis is paramount and frequently dictates the subsequent prognosis of treatment.

A notable strength of the article is its exploration of artificial intelligence's integration into the field of diagnostics, accompanied by a discussion of the associated challenges.  The integration of artificial intelligence diagnostics within this domain is a subject of particular interest, as are the challenges associated with this integration. A thorough exposition of machine learning algorithms and their algorithms is provided, accompanied by a discussion of their primary errors.

The authors have adeptly systematized the existing knowledge on this subject by conducting a meticulous literature review.

This article will undoubtedly be of interest to researchers involved in this field of medicine

Minor remarks:

The table structure does not adequately emphasize the names of columns such as "Method Name" and "Main Algorithms Used."

Author Response

Dear Reviewer 1,

Many thanks for your time and for your positive feedback on this critical paper! Your comment was included in this revision and we hope you like this edited version!

Reviewer 2 Report

Comments and Suggestions for Authors

Title

The title is informative but could be more concise. Consider rewording for clarity, e.g., "Artificial Intelligence in Medical Diagnostic Biosensors: Challenges and False Results."

Abstract

  • The phrase "the first critical state-of-the-art review article" is a bold claim. Ensure there is evidence that no similar review exists.
  • The use of "illustrated reasons" is vague, suggest adding detail explanations?
  • The sentence "expert opinion with further future perspectives is presented..." should clarify whether these perspectives are based on new research findings or general expert insights.
  1. Introduction
  • The sentence "Such devices utilize biological recognition elements like certain proteins..." lacks citations for credibility.
  • The term "give birth to measurable signals" is informal; consider replacing it with "generate measurable signals."
  • The transition from biosensors to AI is abrupt. A brief explanation of AI’s role in biosensing before discussing its integration would improve coherence.

  1. The Concept of "Biosensor" in Medicine

2.1 The Concept of "Sensor"

  • Provide examples of each sensor type with specific real-world applications
  • The description of passive and active sensors could benefit from an example. Add a brief introduction explaining why sensors are crucial in diagnostics
  • The section is overly technical without introductory explanations for non-specialist readers. Simplify technical explanations where possible.

4.2 The Common "Biosensor"

  • Line 99: The sentence "it is composed of the following five components..." should be introduced with a brief explanation of why these components are crucial. Introduce the importance of biosensor components before listing them.
  • The section on biosensor generations is well-written but could benefit from a summary table to enhance readability and comparison for each biosensor generations.

2.3. Biosensors with Integrated Artificial Intelligence

  • Line 216: The phrase "Artificial Intelligence for the Next 20 Years Research Roadmap" requires adding reference citation.
  • The section jumps between AI history and its application in biosensors without a clear structure. Improve the organization by separating AI history from AI applications in biosensors.
  • Line 263: The list format in "AI-driven data processing is divided into four stages..." is beneficial but lacks real-world examples.

  1. Current Common Types of Diagnostic Biosensors and False Results
  • The term "false results" is used repeatedly but should be more clearly defined (e.g., misclassification, incorrect readings, etc.).
  • Some explanations are too technical and could use practical examples by providing case studies or examples of diagnostic errors.
  • The section lacks a summary of key challenges for each biosensor type.

3.1.1. Enzyme-based Biosensors

  • Issues:
  • The list format makes it difficult to grasp key differences between biosensor generations. Use a table to compare biosensor generations.
  • Line 351: The phrase "the amperometric enzyme-based biosensors..." should specify why amperometry is a preferred technique.
  • Line 524: The explanation of "cross-talk in enzyme biosensors" is interesting but lacks experimental evidence.

3.1.2 Immunosensors

  • Strengths: Good breakdown of ELISA and lateral flow immunoassays.
  • The section is too focused on technique descriptions; more emphasis on false positives/negatives is needed.
  • Line 676: The phrase "pregnancy and multiparous women..." in discussing false positives is unclear—does it mean hormone fluctuations affect test accuracy? Clarify the link between pregnancy and false positives.
  • Line627: The section on Point-of-Care testing lacks specific examples from recent studies. Include recent examples of PoC biosensor failures.

3.1.3. Whole-Cell Based Biosensors

  • Strengths: Covers applications in environmental and medical fields.
  • Line 710: The phrase "they lack the selectivity of enzyme-based biosensors" should be supported with examples. Provide specific examples of low selectivity issues.
  • Line 770: The discussion of genetic modifications lacks clarity—how do mutations impact biosensor reliability?

3.1.4 Nucleic Acid-based Biosensors

  • Line 830: The phrase "Nucleic acid-based biosensors have garnered increasing attention due to their sequence variability and specificity." is vague. What type of sequence variability? Be specific.
  • Define the types of nucleic acid biosensors (e.g., DNA, RNA, aptamer-based) before diving into hybridization.
  • Line 861: The section should provide real-world examples of nucleic acid biosensors used in diagnostics (e.g., PCR-based biosensors for COVID-19).
  • Mention commercial applications or FDA-approved biosensors to strengthen relevance.
  • The section discusses the hybridization process but does not sufficiently address false positives/negatives.
  • Line 943: Add a subsection explaining why hybridization errors occur (e.g., non-specific binding, temperature variations, sequence mismatches).
  • Line 955 and onward suggest to add in a new Section 3.1.5 Aptamer-based Biosensors. The section highlights the benefits but does not critically compare aptamer-based biosensors to antibody-based biosensors. Add a table comparing aptamers vs. antibodies in biosensing.
  • The discussion lacks limitations, such as: Stability in different pH levels, Sensitivity to nuclease degradation, Cost of aptamer synthesis, False Results Section Missing.
  • Unclear Figure 1.

3.2. Current Common Biosensors Based on Their Transducers - Detection System

3.2.1. Electrochemical Biosensors

    • The section is detailed but lacks recent advancements, such as nanomaterial-based transducers.
    • Line 1016 :The explanation of voltammetric biosensors should clarify differences between square-wave, cyclic, and differential pulse voltammetry. Figures or diagrams would enhance comprehension.

3.2.2. Optical Biosensors

  • Section 3.2.2.1: Some explanations, such as colorimetric biosensors, assume prior knowledge. Briefly explaining color change principles would improve accessibility.
  • Section 3.2.2.2: Fluorescent biosensors: The limitations due to photobleaching and quenching should be further elaborated.

3.2.3 Thermal Biosensors 

  • The impact of environmental conditions on thermal stability should be discussed.

3.2.4. Acoustic Biosensors 

  • More examples of real-world applications would strengthen this section.
  • Line 1495: The challenges in surface functionalization could be briefly mentioned.

3.3. Current Common Biosensors Based on Their Technology

  • SPR Biosensors
  • More focus on real-time applications in clinical diagnostics.
  • Line 1618: Discuss recent innovations in plasmonic nanostructures.
    • Microfluidic Biosensors
  • A brief comparison of paper-based microfluidics vs. chip-based microfluidics would be useful.
  • The potential limitations of microfluidic systems should be addressed.
  1. Biosensor Performance Factors
  • More emphasis on reproducibility challenges, particularly in point-of-care applications.
  • Stability: The impact of long-term storage conditions should be mentioned.
  1. Future Prospects and Challenges
  • More discussion on AI-based biosensor data interpretation.
  • Address regulatory hurdles and standardization challenges.
  1. Conclusion
  • End with a stronger statement about the transformative potential of biosensors.
  • Minor sentence restructuring would improve the readability of the final paragraph.
Comments on the Quality of English Language

The paper demonstrates a strong command of technical terminology, but there are multiple areas where the English quality could be improved. Below are the key issues identified:

  1. Grammar and Syntax Issues
  • Sentence structure: Many sentences are overly long and complex, making them difficult to follow. Breaking them into shorter, clearer sentences would improve readability.
  • Redundancy: There are instances where the same idea is repeated in different ways, which makes the text unnecessarily long.
  • Wordiness: Some phrases could be more concise. For example, instead of "give birth to measurable signals," a clearer phrase would be "generate measurable signals."
  1. Formality and Academic Style
  • Inconsistent use of formal language: While most of the paper maintains an academic tone, some phrases are too informal, such as "give birth to measurable signals" or "AI-powered biosensors will dominate medical diagnostics."
  • Overuse of absolute statements: The claim that this is the "first critical state-of-the-art review" is too strong and may require justification.
  • Repetitive phrasing: The term "false results" is used repeatedly without clear differentiation between false positives, false negatives, and misclassification errors.
  1. Clarity and Readability
  • Transitions between topics: Some sections, especially those discussing AI and biosensors, shift between ideas without clear transitions.
  • Unclear definitions: Terms like "illustrated reasons" and "expert opinion" should be defined more explicitly.
  • Technical jargon without explanation: The paper assumes the reader is familiar with complex concepts like "Machine Learning (ML), natural language processing, and computer vision" without providing introductory explanations.
  1. Punctuation and Typographical Errors
  • Inconsistent spacing and formatting: Some sections lack proper paragraph breaks, making them visually dense.
  • Comma splices and run-on sentences: Some long sentences should be split to enhance clarity.
  • Unnecessary capitalization: Words like "Artificial Intelligence (AI) Biosensors" do not always need capitalization unless they are in a title or heading.
  1. Suggestions for Improvement
  • Sentence simplification: Break long sentences into two or more for better readability.
  • Word choice refinement: Use more precise terms to avoid vagueness.
  • Grammar check and proofreading: A professional proofreader or AI grammar tool (e.g., Grammarly) can help correct minor errors.

Better transitions: Improve flow between sections with linking sentences

Author Response

Comments and Suggestions for Authors

Title

The title is informative but could be more concise. Consider rewording for clarity, e.g., "Artificial Intelligence in Medical Diagnostic Biosensors: Challenges and False Results."

Dear Reviewer 2, many thanks for your valuable time in this paper and for your thorough comments on it, we appreciate your help so as to make this paper much better than it really was. Title was altered according to your wishes. However, we could not remove technical words since this topic by itself is so technical, hope you understand.

Abstract

  • The phrase "the first critical state-of-the-art review article" is a bold claim. Ensure there is evidence that no similar review exists.

We have ensured that there exists no similar review and that is why we have used this phrase, so as to highlight our work in the field.

  • The use of "illustrated reasons" is vague, suggest adding detail explanations?

We have added a sentence according to the other reviewer and also we have altered this phrase according to you.

  • The sentence "expert opinion with further future perspectives is presented..." should clarify whether these perspectives are based on new research findings or general expert insights.

It has now been clarified, thank you

  1. Introduction
  • The sentence "Such devices utilize biological recognition elements like certain proteins..." lacks citations for credibility.

This sentence has a clear citation, the reference 1

  • The term "give birth to measurable signals" is informal; consider replacing it with "generate measurable signals."

It’s been altered, many thanks

  • The transition from biosensors to AI is abrupt. A brief explanation of AI’s role in biosensing before discussing its integration would improve coherence.

This point has been edited according to your comment

  1. The Concept of "Biosensor" in Medicine

2.1 The Concept of "Sensor"

  • Provide examples of each sensor type with specific real-world applications

We have used the appropriate reference in order to provide some real world results of these types of sensors, because the concept of this paper is the false results thus we have not analyzed thoroughly all the initial theory, but we had to include these basics so as for our paper to cover the topic from its beginning

  • The description of passive and active sensors could benefit from an example. Add a brief introduction explaining why sensors are crucial in diagnostics

A sentence was added in the beginning paragraph of sensors

  • The section is overly technical without introductory explanations for non-specialist readers. Simplify technical explanations where possible.

Technical explanations were simplified where possible

4.2 The Common "Biosensor"

  • Line 99: The sentence "it is composed of the following five components..." should be introduced with a brief explanation of why these components are crucial. Introduce the importance of biosensor components before listing them.
  • The section on biosensor generations is well-written but could benefit from a summary table to enhance readability and comparison for each biosensor generations.

The review focuses on false results and its figures/tables and not in the initial theories, however this table is now added according to you, thanks

2.3. Biosensors with Integrated Artificial Intelligence

  • Line 216: The phrase "Artificial Intelligence for the Next 20 Years Research Roadmap" requires adding reference citation.

We had used this reference for the whole paragraph, so now we edited it in order to be precise.

  • The section jumps between AI history and its application in biosensors without a clear structure. Improve the organization by separating AI history from AI applications in biosensors.

This section is now divided into 2 hyposections according to your comments

  • Line 263: The list format in "AI-driven data processing is divided into four stages..." is beneficial but lacks real-world examples.

We have added an example for each one based on its reference

  1. Current Common Types of Diagnostic Biosensors and False Results
  • The term "false results" is used repeatedly but should be more clearly defined (e.g., misclassification, incorrect readings, etc.)

Where possible this term was altered, but the main focus of the text is the final output which is the false result, also, all reasons for false results are presented clearly, so one is able to see if it os a misclassification or an incorrect reading

  • Some explanations are too technical and could use practical examples by providing case studies or examples of diagnostic errors

In this section, which is mainly technical, we tried to use as much practical infos as possible

  • The section lacks a summary of key challenges for each biosensor type.

We have made the summary on the basic challenges in the figure 1, as in the other sections there is a summary with a table or a figure as well

3.1.1. Enzyme-based Biosensors

  • Issues:
  • The list format makes it difficult to grasp key differences between biosensor generations. Use a table in biosensor generations, so one can easily now see the challenges of each generation by the challenges of their examples

We have added the Table1 for generations and comparison

  • Line 351: The phrase "the amperometric enzyme-based biosensors..." should specify why amperometry is a preferred technique.

This sentence was clarified in the section for the amperometric biosensors

  • Line 524: The explanation of "cross-talk in enzyme biosensors" is interesting but lacks experimental evidence.

you’re right but even it has not

3.1.2 Immunosensors

  • Strengths: Good breakdown of ELISA and lateral flow immunoassays.

Thank you

  • The section is too focused on technique descriptions; more emphasis on false positives/negatives is needed.

We had to analyze the basics for technique descriptions, but if we add more text on false results here it will be a very long review, and also,

  • Line 676: The phrase "pregnancy and multiparous women..." in discussing false positives is unclear—does it mean hormone fluctuations affect test accuracy? Clarify the link between pregnancy and false positives.

As you can see, the sentence is on antibodies, so we speak about the extra antibody load in these cases

  • Line627: The section on Point-of-Care testing lacks specific examples from recent studies. Include recent examples of PoC biosensor failures.

In the provided references, there are examples of recent POC biosensor failures in COVID19

3.1.3. Whole-Cell Based Biosensors

  • Strengths: Covers applications in environmental and medical fields.

Thank you very much

  • Line 710: The phrase "they lack the selectivity of enzyme-based biosensors" should be supported with examples. Provide specific examples of low selectivity issues.

This will be discussed in the following sentences and the following hyposection particularly on enzyme-based biosensors

  • Line 770: The discussion of genetic modifications lacks clarity—how do mutations impact biosensor reliability?

Thank you, this is discussed in the last paragraph of this hyposection, so for more infos one can see these references [refs 80, 87]

3.1.4 Nucleic Acid-based Biosensors

  • Line 830: The phrase "Nucleic acid-based biosensors have garnered increasing attention due to their sequence variability and specificity." is vague. What type of sequence variability? Be specific.

This was clarified, thanks

  • Define the types of nucleic acid biosensors (e.g., DNA, RNA, aptamer-based) before diving into hybridization.

Thank you, this was added

  • Line 861: The section should provide real-world examples of nucleic acid biosensors used in diagnostics (e.g., PCR-based biosensors for COVID-19).

Thank you, examples were added

  • Mention commercial applications or FDA-approved biosensors to strengthen relevance.

We feel that if we add commercial applications only in that paragraph, one could say that we try to promote the discussion on these particular, so we have presented a review only on scientific perceptions from a scientific perspective and away from commercial applications

  • The section discusses the hybridization process but does not sufficiently address false positives/negatives

That is, because as in the previous and the whole paper sections, our review design was to discuss on false results at the last paragraph of each one

  • Line 943: Add a subsection explaining why hybridization errors occur (e.g., non-specific binding, temperature variations, sequence mismatches)

This is added in the last paragraph

  • Line 955 and onward suggest to add in a new Section 3.1.5 Aptamer-based Biosensors. The section highlights the benefits but does not critically compare aptamer-based biosensors to antibody-based biosensors. Add a table comparing aptamers vs. antibodies in biosensing.

We have included aptamer-based on nucleic acid hyposection as it is basically a nucleic acid and the division was made based on the biochemical properties of each biochemical molecule, so their comparison is thorough in the figure 1

  • The discussion lacks limitations, such as: Stability in different pH levels, Sensitivity to nuclease degradation, Cost of aptamer synthesis, False Results Section Missing.

You are right and thank you very much, this sentence was added.

  • Unclear Figure 1.

This happened possibly due to PC/ mdpi system errors, we sent the figure again in the editorial office

3.2. Current Common Biosensors Based on Their Transducers - Detection System

3.2.1. Electrochemical Biosensors

    • The section is detailed but lacks recent advancements, such as nanomaterial-based transducers.

We have discussed particularly on nanomaterials in other subsections and paragraphs

    • Line 1016 :The explanation of voltammetric biosensors should clarify differences between square-wave, cyclic, and differential pulse voltammetry. Figures or diagrams would enhance comprehension.

You are right a sentence was added, the aim of our review is to highlight only the reasons for false results in biosensors, that’s why all of our figures and tables are about false results

3.2.2. Optical Biosensors

  • Section 3.2.2.1: Some explanations, such as colorimetric biosensors, assume prior knowledge. Briefly explaining color change principles would improve accessibility.

You are right, this was added, thanks

  • Section 3.2.2.2: Fluorescent biosensors: The limitations due to photobleaching and quenching should be further elaborated.

Τhese were explained, thank you

3.2.3 Thermal Biosensors 

  • The impact of environmental conditions on thermal stability should be discussed.

Thank you, this was added

3.2.4. Acoustic Biosensors 

  • More examples of real-world applications would strengthen this section.

Α sentence was added, thank you

  • Line 1495: The challenges in surface functionalization could be briefly mentioned.

Thank you, a sentence was added.

3.3. Current Common Biosensors Based on Their Technology

  • SPR Biosensors
  • More focus on real-time applications in clinical diagnostics.

Thanks, a sentence was added

  • Line 1618: Discuss recent innovations in plasmonic nanostructures.

Thanks, a sentence was added

    • Microfluidic Biosensors
  • A brief comparison of paper-based microfluidics vs. chip-based microfluidics would be useful.

Thanks, a sentence was added

  • The potential limitations of microfluidic systems should be addressed.

We have analyzed the limitations of these systems on their basis, ie paper-based or chip-based

  1. Biosensor Performance Factors
  • More emphasis on reproducibility challenges, particularly in point-of-care applications.

Thanks, this was already mentioned in some other paragraphs

  • Stability: The impact of long-term storage conditions should be mentioned.

Thanks, this was already mentioned in some other paragraphs

  1. Future Prospects and Challenges
  • More discussion on AI-based biosensor data interpretation.

Thanks, it was added

  • Address regulatory hurdles and standardization challenges.

Thanks, it was added

  1. Conclusion
  • End with a stronger statement about the transformative potential of biosensors.
  • Minor sentence restructuring would improve the readability of the final paragraph.

Thanks, there is now a new conclusion according to your comments

Comments on the Quality of English Language

The paper demonstrates a strong command of technical terminology, but there are multiple areas where the English quality could be improved. Below are the key issues identified:

  1. Grammar and Syntax Issues
  • Sentence structure: Many sentences are overly long and complex, making them difficult to follow. Breaking them into shorter, clearer sentences would improve readability.

You are right, many sentences are now broken into shorter ones for a better readability

  • Redundancy: There are instances where the same idea is repeated in different ways, which makes the text unnecessarily long.

We tried to remove sentences with the same idea now

  • Wordiness: Some phrases could be more concise. For example, instead of "give birth to measurable signals," a clearer phrase would be "generate measurable signals."

Some phrases were now edited so as to be more concise

  1. Formality and Academic Style
  • Inconsistent use of formal language: While most of the paper maintains an academic tone, some phrases are too informal, such as "give birth to measurable signals" or "AI-powered biosensors will dominate medical diagnostics."

Forgive us, but we tried to make a simpler abstract, introduction and conclusion and overcome the issue with the technical text

  • Overuse of absolute statements: The claim that this is the "first critical state-of-the-art review" is too strong and may require justification.

There is no review on false results in biosensors and AI biosensors, not to say for a critical one. If you search for, there is no review like this one.

  • Repetitive phrasing: The term "false results" is used repeatedly without clear differentiation between false positives, false negatives, and misclassification errors.

Some reasons like temperature, antibodies and many many other ones can trigger either false positives or false negatives, that is why we used “false results”. Also, we explain in the text each reason so one can understand if it is misclassification errors or errors in reading or method errors or any errors, but the aim of the review was to highlight the falses as they actually are: “false”, and we also provide their reasons for a reader to understand why they are false

  1. Clarity and Readability
  • Transitions between topics: Some sections, especially those discussing AI and biosensors, shift between ideas without clear transitions.

The expert opinion has no direct structure, but indeed, it discussed our insights on the topic

  • Unclear definitions: Terms like "illustrated reasons" and "expert opinion" should be defined more explicitly.

Thank you, there has been made a better definition of such terms now

  • Technical jargon without explanation: The paper assumes the reader is familiar with complex concepts like "Machine Learning (ML), natural language processing, and computer vision" without providing introductory explanations.

You’ re right, an appropriate sentence was added now.

  1. Punctuation and Typographical Errors
  • Inconsistent spacing and formatting: Some sections lack proper paragraph breaks, making them visually dense.

We have broken the sections you commented on, also we hope now paragraphs are proper

  • Comma splices and run-on sentences: Some long sentences should be split to enhance clarity.

You are right, some long sentences were now edited, so as to be more accurate

  • Unnecessary capitalization: Words like "Artificial Intelligence (AI) Biosensors" do not always need capitalization unless they are in a title or heading.

You’re right, but we used capitals in order to enter the abbrevs

  1. Suggestions for Improvement
  • Sentence simplification: Break long sentences into two or more for better readability.
  • Word choice refinement: Use more precise terms to avoid vagueness.
  • Grammar check and proofreading: A professional proofreader or AI grammar tool (e.g., Grammarly) can help correct minor errors.

Better transitions: Improve flow between sections with linking sentences

We hope now these issues have been revised appropriately

Reviewer 3 Report

Comments and Suggestions for Authors

This review article investigates the integration of biosensors and AI technology, enhancing real-time monitoring of conditions. The authors described what factors would be effective and cause fluctuation or false results on this kind of biosensor. The topic is quite interesting for the readers of the diagnostic journal. I think that this is a good contribution that merits to be published after some revisions.

  • Please revise the manuscript title to improve the coherence.
  • Abstract and keywords require revision. For example, the authors should explain more about false results and how important it is to investigate them.
  • I recommend adding a section "capacitive biosensor" after impedimetric biosensors.

Author Response

Dear Reviewer 3,

Many thanks for your time and for your positive feedback on this critical paper! All reviewers’ comments were included in this paper, and I hope you like it in this version!